# Molecular architecture of the tumor microenvironment caused by *BRCA1* and *BRCA2* somatic mutations in human lung adenocarcinoma

Gaoming Liao[1,2]*[†], Xinbin Yang[1†], Qi Liu[2†], Shufeng Nan[2], Yan Liu[3], Jinwei Li[3], Si Huang[3], Wang Ning[1], Xionghai Qin[4]*, Gang Xu[1]*

[1]Guangzhou Institute of Cancer Research, the Affiliated Cancer Hospital, Guangzhou Medical University, Guangzhou, China; [2]State Key Laboratory of Respiratory Disease, National Clinical Research Center for Respiratory Disease, National Center for Respiratory Medicine, The First Hospital of Guangzhou Medical University, Guangzhou, China; [3]Guangzhou Medical University, Guangzhou, China; [4]Department of Thoracic Surgery, Harbin Medical University Cancer Hospital, Harbin, China

**\*For correspondence:**
lgm179496478@163.com (GL);
qxh9069@163.com (XQ);
xugang1973@126.com (GX)

[†]These authors contributed equally to this work

**Competing interest:** The authors declare that no competing interests exist.

## eLife Assessment

This **important** study investigates the impact of BRCA1/2 mutations on immunotherapy in lung adenocarcinoma using multi-omics approaches. The detailed genetic analysis of two cancer genes (BRCA1 and BRCA2) demonstrated their new roles in causing the tumor microenvironment in lung cancer. The **solid** findings of this study provide an essential foundation for further developing drugs targeting BRCA1/2 in lung cancer therapy.

**Abstract** Homologous recombination repair (HRR) deficiency is associated with improved immunotherapy responses in non-small cell lung cancer (NSCLC) patients. The HRR genes *BRCA1/2* are key regulators of DNA repair, yet their impact on the tumor microenvironment (TME) in lung adenocarcinoma (LUAD) remains unclear. Using single-cell sequencing and multi-omics data, we characterized *BRCA1/2* mutation-associated transcriptional programs, immune cell composition, and functional alterations in T cells, investigating the molecular and immune architecture of BRCA-mutant LUAD patients. *BRCA1/2* mutations were associated with increased genomic instability and poor prognosis in LUAD patients, but predicted better clinical outcomes following immune checkpoint blockade (ICB) treatment. *BRCA1* mutations correlated with an upregulated type I IFN/IFN-γ signature and CD8+ T cell activation. *BRCA2* mutations were associated with alveolar/stress/inflammatory responses and enhanced MHC-II antigen presentation, linked to CD4+ T cell differentiation. Both alterations coincided with reduced CD28 co-stimulation and CTL activity, hinting at immune evasion. We identified two tissue-resident memory T cell (Trm) subsets as predictors of clinical outcomes and ICB response. *BRCA1* mutations were associated with CD8+ Trm expansion, whereas *BRCA2* mutations were linked to tumor CD4+ Trm expansion and peripheral T/NK cell cytotoxicity. Furthermore, a cancer-promoting program activated by *BRCA1* mutation was vulnerable to histone deacetylase inhibitors, which inhibited LUAD tumor growth. This study provides a preliminary characterization of the BRCA-mutant TME in LUAD patients, revealing distinct transcriptional and immune patterns that highlight differences in *BRCA1/2*-associated molecular architecture and offer a framework for improving therapy efficacy in LUAD.

## Introduction

Lung adenocarcinoma (LUAD), the most common histological subtype of non-small cell lung cancer (NSCLC), remains a leading cause of cancer-related mortality worldwide (*Myers and Wallen, 2025*). The genetic landscape of this disease is complex, involving multiple driver mutations such as *EGFR*, *KRAS*, and genomic instability alterations, which collectively contribute to tumor progression and metastasis (*Skoulidis and Heymach, 2019*; *Nguyen et al., 2022*; *Novikov et al., 2021*). The tumor microenvironment (TME) plays a critical role in shaping tumor progression, immune evasion, and therapeutic response, particularly in the context of genetic mutations affecting DNA repair mechanisms (*de Visser and Joyce, 2023*; *Giraldo et al., 2019*). Homologous recombination repair (HRR) deficiency, characterized by mutations in key DNA repair genes, has been increasingly recognized as a determinant of tumor genomic instability in breast cancer, ovarian cancers, and lung cancer (*Hoppe et al., 2018*; *Liao et al., 2021*; *Yan et al., 2023*).

BReast CAncer gene 1 (*BRCA1*) and gene 2 (*BRCA2*) are crucial mediators of genomic stability, and their mutations often lead to increased DNA damage, heightened immune surveillance, and altered tumor-immune interactions (*Samstein et al., 2021*; *Bruand et al., 2021*). Previous studies revealed that mutations in *Brca1* and *Brca2* mediate distinct immune microenvironment effects in murine models of breast cancer, and showed that *Brca2* deficiency improved response to immune checkpoint inhibitors (*Samstein et al., 2021*; *Francis et al., 2015*). Among the key genetic mutations that drive LUAD, *BRCA1* and *BRCA2* mutations (with prevalence rates of approximately 4% and 5%, respectively) have been increasingly implicated in the pathogenesis and progression of lung cancer (*Yan et al., 2023*; *Lee et al., 2020*). Recent studies suggest that HRR-deficient tumors exhibit distinct immunogenic profiles, influencing immune cell infiltration, antigen presentation, and responses to immune checkpoint blockade (ICB) therapy in NSCLC (*Chabanon et al., 2016*; *Zhou et al., 2022*). *BRCA1* mRNA can predict the efficacy of second-line cisplatin chemotherapy in patients with metastatic NSCLC and has a good predictive value for patients with advanced NSCLC receiving platinum-based chemotherapy (*Papadaki et al., 2012*; *Bonanno et al., 2013*). One report showed a case of *BRCA2*-positive lung adenocarcinoma, in which the disease was stable for about 2 years after treatment with olaparib (*Motohashi et al., 2024*). Despite this, the molecular architecture of the TME in LUAD with *BRCA1* and *BRCA2* mutations remains incompletely understood. Understanding the molecular mechanisms through which *BRCA1* and *BRCA2* mutations alter the TME in LUAD is crucial for identifying potential therapeutic targets and improving treatment strategies. To address these challenges, we leveraged single-cell RNA sequencing (scRNA-seq) and single-cell T cell receptor sequencing (scTCR-seq) to comprehensively dissect the *BRCA1/2*-driven TME in LUAD. scRNA-seq enables transcriptomic profiling at the single-cell level, providing unparalleled insights into cellular heterogeneity, lineage differentiation, and gene expression dynamics within tumors (*Jovic et al., 2022*; *Travaglini et al., 2020*). Unlike bulk RNA sequencing, which averages gene expression across heterogeneous cell populations, scRNA-seq allows for the identification of distinct malignant and immune cell subsets, as well as their functional states (*Haque et al., 2017*). Furthermore, scTCR-seq offers a powerful approach for understanding T cell diversity, clonal expansion, and antigen-specific immune responses (*Frank et al., 2023*).

In this study, we leveraged large-scale genomic and transcriptomic datasets to investigate the prognostic implications of *BRCA1/2* mutations in LUAD patients (nearly 2000 samples). By integrating scRNA-seq and scTCR-seq, this study provides a high-resolution characterization of the immune microenvironment and antigen-specific T cell responses in *BRCA1/2*-mutant LUAD. Our findings reveal how *BRCA1/2* mutations shape the immune landscape, influence CD8[+] and CD4[+] T cell populations, and modulate immunotherapy outcomes. These insights offer a comprehensive view of the molecular mechanisms underlying *BRCA1* and *BRCA2* mutations in LUAD and highlight new opportunities for personalized therapeutic strategies.

## Results

### BRCA mutations are associated with genomic instability and poor prognosis in lung adenocarcinoma

Using the TCGA-LUAD cohort data, we examined the effect of somatic mutations of *BRCA1* and *BRCA2* on genomic instability. The results showed that the mutations in both genes could significantly

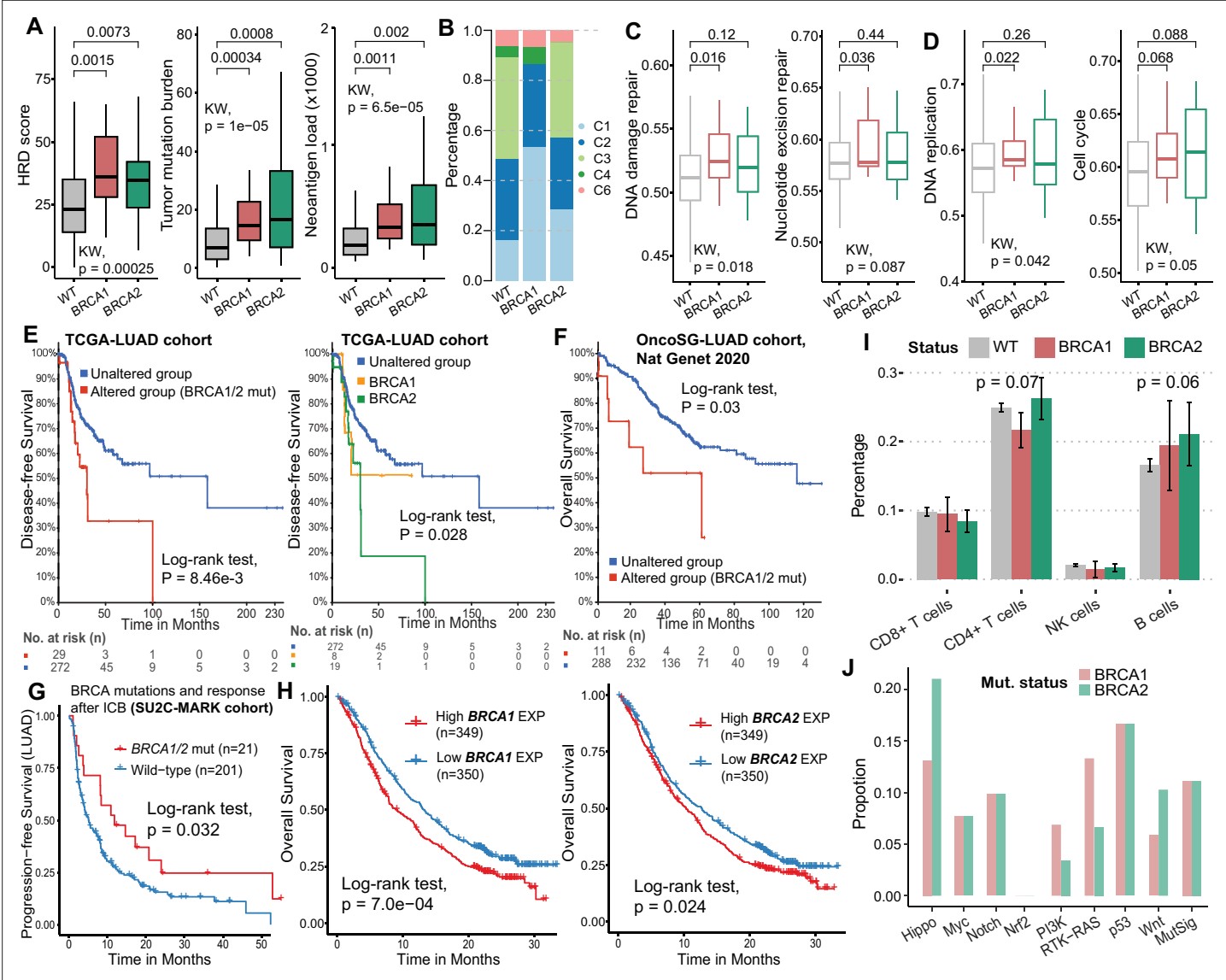

**Figure 1.** *BRCA1* and *BRCA2* mutations are associated with genomic instability and poor prognosis of LUAD. (**A**) Distribution of HRD score, TMB, and neoantigen load in patients with wild-type (n=516), *BRCA1*-mutated (n=19), and *BRCA2*-mutated (n=29) lung adenocarcinoma from the TCGA-LUAD cohort. The significance level was calculated by Kruskal-Wallis test (among the three groups) and two-sided Wilcoxon rank-sum test (between the two groups). (**B**) Proportion of immune subtypes. The significance level was calculated by Chi-squared test. C1: wound healing, C2: IFN-g dominant, C3: inflammatory, C4: lymphocyte depleted, C6: TGF-b dominant. (**C–D**) Activity of DNA damage repair and cell cycle-related pathways in wild-type, *BRCA1*-, and *BRCA2*-mutated patients (TCGA-LUAD cohort). (**E–F**) Survival analysis of BRCA mutations in TCGA-LUAD (**E**) and OncoSG-LUAD (**F**) cohorts. (**G**) Survival analysis of BRCA mutations in LUAD patients after ICB treatment. (**H**) Survival analysis based on *BRCA1* (left) and *BRCA2* (right) expression in patients with ICB treatment. (**I**) Lymphocyte infiltrates in patients with wild-type, *BRCA1*, and *BRCA2* mutations (TCGA-LUAD cohort). Error bars, means ± SEM (Standard error of the mean). The significance level was calculated by Kruskal-Wallis test. (**J**) Mutation frequencies of the oncogenic signaling pathway genes (in-house data).

The online version of this article includes the following figure supplement(s) for figure 1:

**Figure supplement 1.** *BRCA1* and *BRCA2* mutations are associated with genomic instability.

increase the homologous recombination repair deficiency (HRD) score and present a higher tumor mutation burden (TMB) and fraction of genome altered (FGA; *Figure 1A*, *Figure 1—figure supplement 1A*), indicating elevated genomic instability in LUAD tumors with *BRCA1/2* mutations. This phenomenon was also observed in *BRCA1/2* mutated patients with NSCLC (*Figure 1—figure supplement 1B–C*). Furthermore, *BRCA1* and *BRCA2* mutations increased the neoantigen load in LUADs (*Figure 1A*), suggesting that *BRCA1/2*-mutated tumors might be more immunogenic. By analyzing

the proportion of immune subtypes (*Thorsson et al., 2018*), we found that patients with *BRCA1* mutations had a higher proportion of C1 subtype (wound healing; p=5.80e-04, Chi-squared test), while patients with *BRCA2* mutations and wild-type had a higher proportion of C3 subtype (inflammatory; p=6.61e-03, *Figure 1B*). *BRCA1*-mutated patients also showed activated DNA damage repair pathways and exhibited higher DNA replication and cell cycle activity compared to wild-type patients (*Figure 1C–D*), suggesting that *BRCA1* mutations may promote tumor growth through compensatory DNA repair activation—a pattern that may reflect partial HRD rather than complete loss of function.

Next, we explored the impact of *BRCA1/2* mutations on clinical outcomes of LUAD patients and found that these mutations were associated with worse prognosis for disease-free survival (DFS, p=8.46e-03, Log-rank test; *Figure 1E*) and overall survival (OS, p=0.03; *Figure 1F*) in multiple LUAD cohorts. In particular, *BRCA1* and *BRCA2* mutations as risk factors shortened the survival of LUAD patients (p=0.028; *Figure 1E*). Interestingly, LUAD patients with BRCA mutations had significantly longer progression-free survival after ICB treatment (*Figure 1G*). Through transcriptional analysis, we found that upregulation of both *BRCA1* and *BRCA2* was associated with worse prognosis after ICB treatment (p<0.05, *Figure 1H*), consistent with *BRCA1/2* expression as a risk factor for lung cancers (*Yan et al., 2023*). The apparent paradox between favorable ICB outcomes linked to *BRCA1/2* mutations and poorer outcomes associated with high *BRCA1/2* expression may be explained by distinct biological roles—where mutations induce genomic instability and immunogenicity, while high expression reflects proficient DNA repair and tumor cell fitness. These results suggest that BRCA abnormalities may have potential to predict ICB treatment outcomes. In addition, we found the inconsistencies in lymphocyte infiltration between *BRCA1* and *BRCA2* mutations, suggesting heterogeneity in the immune microenvironment (*Figure 1I*).

## Single-cell landscape of BRCA mutations in LUAD patients

Exome sequencing was performed on four samples from two individuals: two tumor tissues and two matched blood samples. The results showed that the two types of tumor tissues carried non-synonymous mutations of driver genes *TP53*, *EGFR*, and *FAT1* (*Supplementary file 1A*), and had similar mutation frequencies in oncogenic signaling pathways, including p53 and Notch pathways, as well as significant mutation gene set (MutSig) scrutinized by MutSigCV (*Sanchez-Vega et al., 2018*; *Figure 1J*). Exome sequencing data show that these two types of tumor tissues harbor somatic nonsynonymous single-nucleotide variants (SNV) in *BRCA2* (p.N372H) and *BRCA1* (p.E991G, p.S1566G, p.K1136R, p.P824L, and p.Y809H), respectively (*Supplementary file 1A*). The *BRCA2* p.N372H variant lies within the BRC3 or BRC4 motifs critical for RAD51 binding. It may alter binding affinity, impair high-fidelity HRR, and promote genomic instability (*Healey et al., 2000*; *Galkin et al., 2005*; *Jimenez-Sainz et al., 2022*). In *BRCA1*, mutations are distributed across two key functional domains: the Coiled-Coil domain (e.g. p.E991G, p.Y809H, p.P824L) and the BRCT domain (e.g. p.K1136R, p.S1566G). Coiled-Coil mutations disrupt BRCA1-PALB2-BRCA2 complex assembly, impairing localization to DNA damage sites and subsequent *RAD51* recruitment; BRCT domain mutations compromise phospho-protein recognition and G2/M checkpoint control, leading to defective DNA damage response and unchecked proliferation of damaged cells (*Adamovich et al., 2022*; *Mangkusaputra et al., 2025*; *Xu et al., 2025*). Together, these defects promote the accumulation of genomic scars and chromosomal instability.

To preliminarily explore the impact of *BRCA1/2* mutations on the immune microenvironment, scRNA-seq was performed on LUAD patients carrying *BRCA1* or *BRCA2* somatic mutations (a total of six samples, including two tumors, two adjacent normal tissues, and two peripheral blood samples from one *BRCA1*-mutant and one *BRCA2*-mutant patient). A total of 69,180 high-quality cells were obtained after QC, of which 27,851 (40.3%) were from tumor tissues, 25,035 (36.2%) were from adjacent normal tissues, and 16,294 (23.5%) were from blood (*Figure 2A*, *Figure 2—figure supplement 1A and B*). Of these cells, 35,450 cells (51.2%) and 33,730 cells (48.8%) were from the *BRCA1*- and *BRCA2*-mutant samples, respectively. After batch correction (Harmony), we performed cell clustering and cataloged all cells into 19 main cell types annotated with canonical marker genes (*Figure 2A-C*, *Figure 2—figure supplement 1A-C*), identifying 4 epithelial compartments (alveolar type 1 [AT1], alveolar type 2 [AT2], ciliated, and club cells), 2 stromal compartments (fibroblasts and endothelial cells), 5 lymphocyte compartments (CD4$^+$ T, CD8$^+$ T, natural killer-NK/NKT, B cells, and plasma), and 6 myeloid compartments (macrophages, dendritic cells-DCs, classical/nonclassical monocytes,

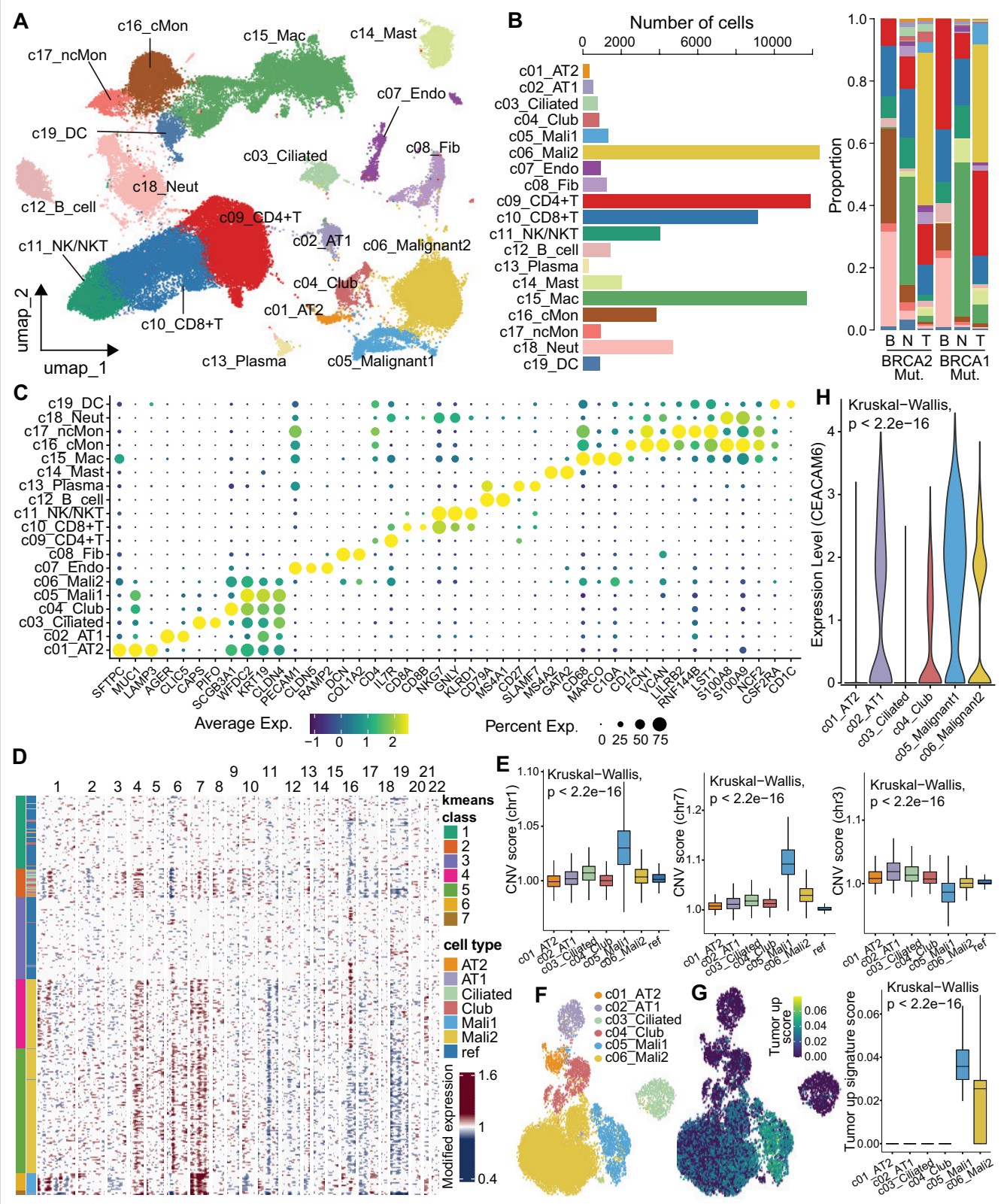

**Figure 2.** Single-cell transcriptome analysis and LUAD malignant cells identification. (**A**) UMAP visualization of cell types in patients with *BRCA1/2* mutations (in-house data). (**B**) The number of each cell type and its proportion in different samples, including tumor tissue (**T**), adjacent normal tissue (**N**), and peripheral blood (**B**). (**C**) The bubble plot shows the expression of the markers used for cell type identification. (**D**) The heat map inferred CNVs in epithelial cells, malignant cells, and reference cells. (**E**) Distribution of CNV scores across different cell types on specific chromosomes. (**F**) UMAP

*Figure 2 continued on next page*

Figure 2 continued

visualization of CNV score in epithelial cells and malignant cells. (**G**) UMAP visualization (left) and box plot (right) of tumor upregulated signature score. (**H**) Expression of *CEACAM6* in epithelial and malignant cells.

The online version of this article includes the following figure supplement(s) for figure 2:

**Figure supplement 1.** Single-cell transcriptome analysis in LUAD patients.

neutrophil, and mast cells). Among immune cells, the most abundant cell types were T lymphocytes and macrophages (**Figure 2B**).

To distinguish malignant cells, we first inferred copy number variation (CNV) of all epithelial cell types in solid tissues using the inferCNV pipeline (**Tirosh et al., 2016**; **Figure 2—figure supplement 1D–E**). Using immune cells as a control and comparing the CNV scores of each epithelial cell subset, we identified two malignant cell subsets that were significantly clustered together using the K-means clustering algorithm and exhibited higher scores of oncogenic CNV in LUAD (**Figure 2D–E**). For example, both chromosome 1q21 region and chromosome 7, frequently amplified in LUAD tumors (**Munkhbaatar et al., 2020**; **Wang et al., 2023**), showed significantly higher CNV scores in the malignant cells of this study (**Figure 2D–E**). The previously reported loss of chromosome 3q (**Karasaki et al., 2023**) was also observed in malignant cells (**Figure 2D–E**). To demonstrate the properties of malignant cells, we used bulk RNA-seq data from the TCGA-LUAD cohort to identify tumor upregulated and downregulated gene signatures, and calculated module scores in different epithelial cell types. The malignant cell subsets had significantly higher tumor upregulated signature scores and lower downregulated signature scores (**Figure 2F-G**, **Figure 2—figure supplement 1F**). In addition, tumor marker genes, such as *CEACAM6*, *CLDN4*, and *WFDC2*, showed higher expression in malignant cells (**Figure 2H**, **Figure 2—figure supplement 1C**).

## Distinct malignant transcriptional programs associated with *BRCA1* and *BRCA2* mutations in LUAD

We have identified two subsets of malignant cells (13,694 cells in total) from the *BRCA1/2*-mutant samples. We then asked which key molecules were specifically activated in malignant cell subsets and whether there were differences in the oncogenic programs between the *BRCA1*- and *BRCA2*-mutant cases. To address this, we first performed differential expression analysis comparing malignant cell subsets versus other epithelial cells. The results showed that tumor markers *WFDC2* and *CLDN4* were significantly upregulated in the malignant1 subset, while *CEACAM6* was significantly upregulated in both malignant cell types (**Figure 3—figure supplement 1A** and **Supplementary file 1B**). Previous studies show that *DUSP1* could promote angiogenesis, invasion, and metastasis in NSCLC, and *FTH1* expression protects cells from cell death induced by the GPX4 inhibitor RSL3 (**Moncho-Amor et al., 2011**; **Bayır et al., 2023**). These genes upregulated in malignant cells are often also highly expressed in tumor tissues relative to paracancerous tissue (**Figure 3—figure supplement 1B**). The upregulated genes in malignant1 subset were mainly enriched in ribonucleotide biosynthetic and nucleoside-related metabolic process (**Figure 3A** and **Supplementary file 1C**), suggesting enhanced DNA replication and growth abilities. In contrast, the upregulated genes of malignant2 were mainly enriched in cell adhesion, antigen processing and presentation of peptide antigen, and MHC protein complex assembly (**Figure 3A** and **Supplementary file 1C**), suggesting involvement in lymphocyte recruitment. These patterns were also observed in direct comparison between the two malignant subsets (**Figure 3—figure supplement 2A** and **Supplementary file 1D-E**).

We then applied the nonnegative matrix factorization (NMF) algorithm to identify transcriptional programs consisting of co-expressed genes across all malignant cells. Using this method, we extracted a total of four programs and identified two meta-programs (MP1, MP2) based on program similarities (**Figure 3B**). The MP1 module contains 156 genes (**Supplementary file 1F**), whose activity was significantly higher in *BRCA2*-mutant tumor cells (**Figure 3C**) and was associated with better prognosis in LUAD patients (p=0.031, **Figure 3D**). The favorable prognosis linked to MP1 was related to its involvement in lymphocyte migration, T-cell chemotaxis, and myeloid leukocyte differentiation (**Figure 3E** and **Supplementary file 1H**). The MP2 module contains 200 genes involved in functions such as ATP synthesis/metabolic process and mitochondrial translation, and its activity was higher in *BRCA1*-mutant tumors (**Figure 3C and E**, and **Supplementary**

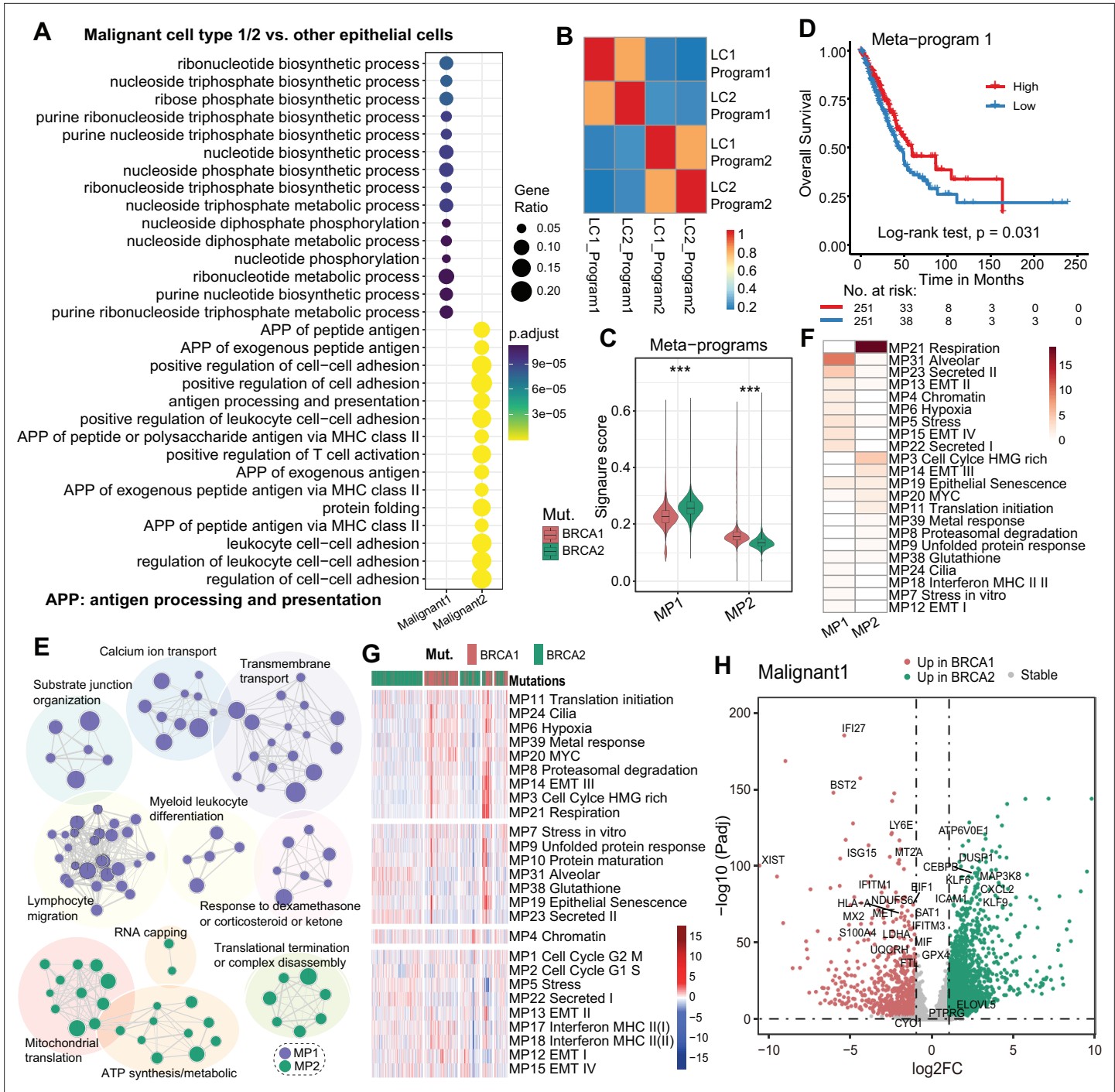

**Figure 3.** Identification of key programs in *BRCA1/2* mutated tumors. (**A**) Functional enrichment analysis of genes that were upregulated in malignant cells relative to normal epithelial cells. (**B**) The heat map of the similarity between pairs of programs is identified based on the NMF algorithm. (**C**) MP score between *BRCA1* and *BRCA2* mutated malignant cells. (**D**) K-M curves for MP1 score survival analysis. (**E**) Enrichment map of the functions enriched by MP1 and MP2 genes. (**F**) The overlap of in-house MPs and known cancer MPs. HMG: high mobility group. (**G**) The heat map shows the activity of known cancer MPs in *BRCA1* and *BRCA2* mutated malignant cells. (**H**) Volcano diagram of differentially expressed genes between *BRCA1* and *BRCA2* mutated malignant cells.

The online version of this article includes the following figure supplement(s) for figure 3:

**Figure supplement 1.** The expression of differential genes.

**Figure supplement 2.** Key programs driven by *BRCA1* and *BRCA2* mutations.

*file 1F-G*). To further contextualize MP1 and MP2, we obtained 41 known cancer MPs defined by *Gavish et al., 2023*. Among these, 15 MPs irrelevant to LUAD were excluded and 26 MPs were included. Comparing our MPs with known MPs, we found that MP1 was mainly enriched in alveolar, stress, and secretion-related MPs, whereas MP2 was enriched with cell-cycle-related genes encoding HMG (high mobility group)-box proteins and with MYC-related MPs (*Figure 3F*). Furthermore, *BRCA1* mutations were associated with hypoxia and oncogenic MYC-related MPs, while *BRCA2* mutations were linked to alveolar, stress, and secretion-related MPs (*Figure 3G*, *Figure 3—figure supplement 2B*). Comparing key molecules, *BRCA1* mutation was associated with upregulation of type I IFN genes (e.g. *IFI27, IFITM3*), angiogenesis genes (*S100A4, S100A10*), and *LDHA* (*Figure 3H*, *Figure 3—figure supplement 2C*, and *Supplementary file 1H*). *BRCA2* mutation was associated with upregulation of stress-related genes (e.g. *GADD45B, DUSP1*, and *SOD2*), immune cell-related genes (*NFKBIA, ICAM1*), inflammatory-related genes (*TNFAIP3, CEBPB*, and *CXCL2*), and apoptosis gene *CFLAR* (*Figure 3H*, *Figure 3—figure supplement 2C*, and *Supplementary file 1H*).

## Inferred transcriptional trajectories from epithelial to malignant cells in *BRCA1*- and *BRCA2*-mutant tissues

Tumor evolution is a process of gradual accumulation of somatic mutations from normal cells (*Salcedo et al., 2025*). To infer potential transcriptional transitions during malignant transformation, we analyzed the transcriptional trajectories using Monocle 2 (*Cao et al., 2019*), revealing a dynamic transitional spectrum from AT1/2 cells to malignant cells (*Figure 4A*, *Figure 4—figure supplement 1A*). The early pseudotime was dominated by AT1 and AT2 cells, primarily from tumor-adjacent and *BRCA2*-mutant tumor tissues (*Figure 4A*). The trajectory then extended into two branches: one dominated by malignant1 subset (87.6% from *BRCA1*-mutant tumor), and the other almost all malignant2 subset, allowing us to explore expression gradients associated with *BRCA1* and *BRCA2* mutations (*Figure 4A*). Branch fate determination gene analysis showed that the malignant1 branch upregulated genes related to oxidative phosphorylation, ATP synthesis and metabolism, and ROS response (*Figure 4—figure supplement 1B*). The malignant2 branch upregulated immune-related genes such as *IL1B*, *TNFAIP3*, and *CD74*, enriched in cytokine-mediated signaling, antigen processing and presentation, and T cell activation and differentiation (*Figure 4—figure supplement 1B*). This suggests heterogeneous transcriptional fates during the transition from normal lung epithelial cells to malignant cells in these two cases. We further investigated the expression differences along the inferred trajectories. The *BRCA1* mutation branch showed activation of innate immune response-related pathways such as type I IFN production, response to IFN-α/β, and upregulated genes related to T cell homeostasis/migration and T cell receptor signaling (*Figure 4B*), suggesting an association with innate immune response and T cell activation. The *BRCA2* mutation branch showed activation of cell growth and development, response to wounding, and wound healing-related functions, along with promotion of MHC protein complex assembly and lymphocyte-mediated immunity (*Figure 4B*), suggesting that *BRCA2* mutation may be associated with immune engagement alongside tumor growth.

Based on pseudotime order, we divided trajectories into 10 bins and analyze the activity changes of related features. The module scores of tumor downregulated genes gradually decreased along pseudotime (*Figure 4C*), while tumor upregulated genes increased (*Figure 4—figure supplement 1C*). Cancer MPs, such as cell cycle G2/M (MP1) and stress (MP5) increased along pseudotime in both *BRCA1*- and *BRCA2*-mutant tissues (*Figure 4D–E*). Density analysis suggested that stress activation was more prominent in *BRCA2*-mutant tumor cells, while cell cycle HMG-rich program was mainly activated in the *BRCA1*-mutant group (*Figure 4F*). Differences in expression programs between *BRCA1* and *BRCA2* mutations were mainly apparent in mid-to-late pseudotime. For example, *BRCA1* mutation-associated fate-determination functions such as response to IFN-α and type I IFN showed higher activity in mid-to-late pseudotime bins (*Figure 4G and H*, *Figure 4—figure supplement 1D*). This pattern was also observed in cGAS-STING innate immune sensing pathways, such as cGAS and IFN-γ response (*Figure 4H*, *Figure 4—figure supplement 1E*). In contrast, TGF-β signaling and stress modules were more active in late pseudotime in *BRCA2*-mutant cells (*Figure 4I*). MHC class I and II molecules showed increased activity in late pseudotime in *BRCA1*- and *BRCA2*-mutant cells, respectively (*Figure 4G–I*). This pattern was also reflected in the cell density analysis (*Figure 4J*). Furthermore, CD8[+] Tcm and Th1 signatures exhibited higher activity in late pseudotime in *BRCA1*- and

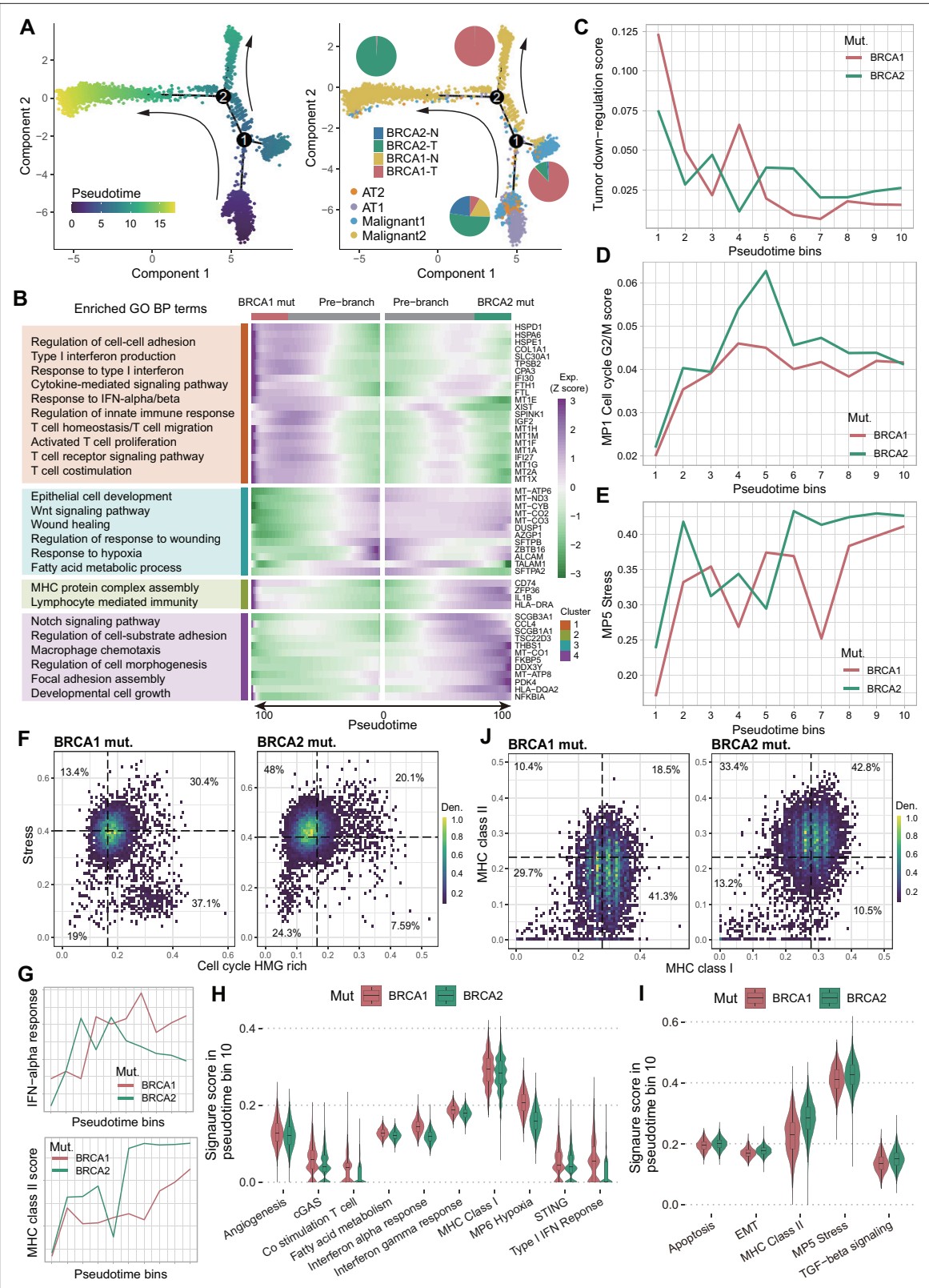

**Figure 4.** Tumor evolutionary analysis of *BRCA1* and *BRCA2* mutations. (**A**) Pseudotime analysis of epithelial and malignant cells. The branched trajectory was colored by pseudotime (left) and cell types (right). The pie chart shows the proportion of sample groups in each branch. (**B**) The top 50 cell fate genes and their enriched GO BP terms of *BRCA1/2* mutations in malignant2 cell type. (**C–E**) Pseudotime was broken down into 10 bins to smooth gene expression patterns. Average gene module score between *BRCA1* and *BRCA2* mutation groups for tumor down-regulation (**C**), MP1

*Figure 4 continued on next page*

Figure 4 continued

cell cycle G2/M (**D**), and MP5 stress (**E**) gene modules. (**F, J**) Cell density distribution based on cell cycle HMG rich versus stress gene modules (**F**) and MHC class I versus MHC class II gene modules (**J**) in the same cells between *BRCA1* and *BRCA2* mutations. Color represents the density of cells. The dotted line represents the median value of the corresponding module score. (**G**) Average gene module score for IFN-α response (top) and MHC class II (bottom) gene signatures. (**H–I**) The activities of representative signatures between *BRCA1* and *BRCA2* mutations in bin 10 of malignant cells according to pseudotime.

The online version of this article includes the following figure supplement(s) for figure 4:

**Figure supplement 1.** Evolutionary analysis of *BRCA1* and *BRCA2* mutant tumors.

*BRCA2*-mutant cells, respectively (*Figure 4—figure supplement 1F–G*). These findings suggest a differential association with CD8⁺ versus CD4⁺ T cell engagement.

## *BRCA1* and *BRCA2* mutations reshape distinct immune microenvironment in LUAD tumors

We next explored how *BRCA1/2* mutations reshape tumorimmune microenvironment of *BRCA2* mutation-upregulated genes was enriched in MHC class II-related antigen presentation and lymphocyte-mediated immunity functions (*Figure 5A*). GSEA showed that MHC class II molecules were upregulated in the *BRCA2* mutation group, while MHC class I molecules were upregulated in the *BRCA1* mutation group (*Figure 5B–C*), consistent with cell density distribution (*Figure 4J*). Similarly, response to IFN-α/γ and type I IFN-related genes, as well as TNF-α family members and receptors, were enriched in *BRCA1*-mutant tumor cells (*Figure 5B and C*, *Figure 5—figure supplement 1A and B*). Similar patterns were observed for DNA repair, fatty acid metabolism, and CD8⁺ T cell signatures (*Figure 5B and D*, *Figure 5—figure supplement 1A*). DNA double-strand breaks (DSBs) can lead to the release of cytoplasmic dsDNA, which can be sensed by cGAS-STING signaling to trigger innate immune responses and type I IFN production, promoting anti-tumor immunity (*Harding et al., 2017*; *Kwon and Bakhoum, 2020*; *Chen et al., 2016*). Based on these observations, we hypothesized that *BRCA1* mutation promotes DSBs and triggers the activation of cGAS-STING signaling as well as CD8⁺ T cell infiltration. Indeed, *BRCA1* mutations upregulated DNA damage-related checkpoints and significantly activated DSBs-related DNA repair pathways, including DNA double-strand break repair, homology-directed repair, and HRR (*Figure 5E and F*, *Figure 5—figure supplement 1C and D*). Furthermore, our results revealed that *BRCA1*-mutant tumors showed higher activity of cGAS-STING signaling and STING-mediated induction of host immune responses compared to *BRCA2*-mutant tumors (*Figure 5G*, *Figure 5—figure supplement 1F*). Also, cGAS-STING signaling genes, including *cGAS*, *STING1*, and downstream factors *STAT1* and *CCL5*, were upregulated in *BRCA1*-mutant tumor cells (*Figure 5H*). This observation was validated through immunofluorescence staining experiments on patient tumor tissue sections (*Figure 5I–J*). Immune activity analysis suggested that *BRCA1* mutation was associated with enhanced activity of CD8⁺ T cells and myeloid cells (*Figure 5—figure supplement 1G*), consistent with GSEA results. Notably, the increased infiltration of CD8⁺ T cells associated with *BRCA1* mutations was also confirmed in lymphocyte subsets (*Figure 5K*).

Unlike *BRCA1* mutation, upregulated genes in *BRCA2*-mutant tumor cells were enriched in inflammatory response, interleukin receptors, and chemokines (*Figure 5B*, *Figure 5—figure supplement 1A*), which was confirmed by cell activity analysis. For example, the IL2/IL17 pathway, 4-1BB pathway, and cytokine pathway were more active in *BRCA2*-mutant tumor cells (*Figure 5E*, *Figure 5—figure supplement 1C*), consistent with the higher proportion of inflammatory subtypes in *BRCA2*-mutant patients (*Figure 1B*). In addition, *BRCA2* mutation was also associated with activation and differentiation of CD4⁺ T cells (*Figure 5B and K*). For example, *BRCA2*-upregulated genes were enriched in CD4⁺ Tem signature and showed higher activity of Th1/Th2 pathways, Th1 toxicity, and MHC class II modules (*Figure 5B-E*, *Figure 5—figure supplement 1C*). Indeed, immune activity analysis also suggested that *BRCA2*-mutant tumor cells were more associated with the activation of Th1, Th2, and Treg cells compared to *BRCA1* mutations (*Figure 5—figure supplement 1G*). Together, these observations suggested that *BRCA1* mutations are associated with cGAS-STING-mediated host immune responses and CD8⁺ T cell infiltration, while *BRCA2* mutations are linked to MHC II molecule-mediated antigen presentation, inflammatory responses, and CD4⁺ T cell differentiation.

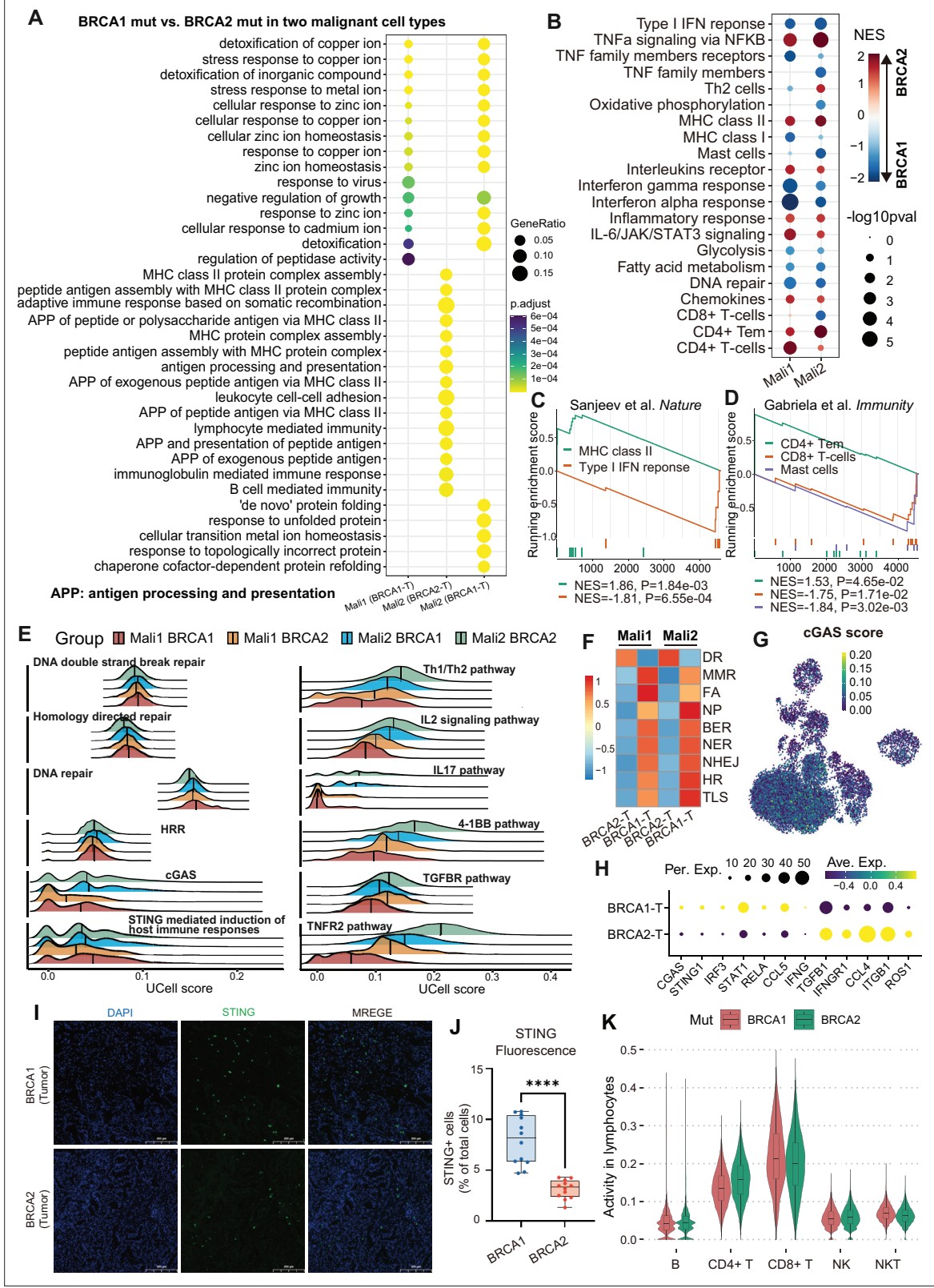

**Figure 5.** *BRCA1/2* mutations are associated with tumor lymphocyte activation. (**A**) Functional enrichment of differentially expressed genes in malignant cells between *BRCA1* and *BRCA2* mutation groups. (**B**) GESA of immune response profile between *BRCA1* and *BRCA2* mutations. Red (blue) indicates the gene set enriched in *BRCA2* (*BRCA1*)-mutated malignant cells. (**C–D**) GESA plot of immune signature (**C**) and gene module of immune cell type (**D**) in malignant2 subset. NES: normalized enrichment score. (**E**) Density ridge plot of representative pathways in two types of malignant subsets with

*Figure 5 continued on next page*

*Figure 5 continued*

*BRCA1* and *BRCA2* mutations. HRR: homologous recombination repair. (**F**) Mean of pathway activities related to DNA damage repair across two types of malignant cells with *BRCA1* and *BRCA2* mutations. (**G**) UMAP visualization of cGAS pathway score. (**H**) The expression of the representative markers. (**I**) Immunofluorescence staining for STING (green) and DAPI (blue) on lung tumor sections from patients with *BRCA1* and *BRCA2* mutations. Scale bars, 200 μm. (**J**) Quantification and estimation of the relative intensity of the STING⁺ cells. To compare two groups, the p value was computed with the two-sided Welch's t test (ns, p>0.05; *, p<0.05; **, p<0.01; ***, p<0.001; ****, p<0.0001). (**K**) Lymphocyte activity between *BRCA1* and *BRCA2* mutations in lymphocyte cells.

The online version of this article includes the following figure supplement(s) for figure 5:

**Figure supplement 1.** Heterogeneity of *BRCA1/2* mutations in tumor lymphoid activity.

## *BRCA1* and *BRCA2* mutations facilitate the clonal expansion of tissue-resident memory T cells

To examine the effect of *BRCA1/2* mutations on T lymphocytes, we obtained T cell data from BRCA wild-type in LUAD as controls from a previous study (*Deng et al., 2024*) and combined it with our T cell data for integrated analysis. Nine T cell subsets (67,823 cells) were identified (*Figure 6A*, *Figure 6—figure supplement 1A-C*). Naive T cells were mainly distributed in blood, while Teff cells were more abundant in blood and paracancerous tissues than in tumors (*Figure 6B*). Treg cells were decreased in *BRCA1/2* mutant tumors compared to wild-type (*Figure 6—figure supplement 1D*). We identified two tissue-resident memory T cell (Trm) subsets, CD8⁺ Trm and CD4⁺ Trm, both predominantly derived from tumor tissues (*Figure 6B*). Interestingly, our analysis revealed that CD8⁺ Trm cells were more abundant in *BRCA1*-mutant tumors, whereas CD4⁺ Trm cells were more abundant in *BRCA2*-mutant tumors (*Figure 6B-D*, *Figure 6—figure supplement 1D*, and *Figure 6—figure supplement 2A–B*). CD8⁺ Trm cells highly expressed TNF family members and showed high activity for interferons and chemokines, similar to Teff and NK/NKT cells, whereas CD4⁺ Trm cells upregulated chemokine receptors and cytokine receptors (*Figure 6—figure supplement 1E*). Trm score of both subsets could predict the overall survival of NSCLC patients after ICB treatment (p<0.01, *Figure 6—figure supplement 2C–D*) and were associated with ICB response (p<0.05, *Figure 6—figure supplement 2E*), suggesting Trm cells as potential biomarkers for ICB treatment in lung cancer.

Consistent with malignant cell results, *BRCA1* mutation was associated with upregulation of TNF family members in T lymphocytes compared to *BRCA2* mutation and wild type (*Figure 6—figure supplement 1E-G* and *Figure 6—figure supplement 3*). Chemokine receptor and TGF-β were more active in *BRCA2*-mutant tumor tissues (*Figure 6—figure supplement 1G-I* and *Figure 6—figure supplement 3A*). *BRCA1/2* mutations were associated with enhanced antigen presentation and cytokine signaling compared to wild type (*Figure 6—figure supplement 1H-J* and *Figure 6—figure supplement 3A*). Using tissue-specific T-cell signatures (*Wienke et al., 2024*), we calculated activity at the single-cell level (*Figure 6E*, *Figure 6—figure supplement 4A*), and found that *BRCA1/2* mutations were associated with reduced CD28 family co-stimulation and decreased cytotoxic T lymphocytes (CTLs) and NKT cell infiltration compared to wild type (*Figure 6E–F*), which explains why patients with *BRCA1/2* mutations showed worse prognosis (*Figure 1E–F*). *BRCA1* mutations were associated with reduced TIL dysfunction and exhaustion dysfunction in tumors (*Figure 6E–F*), consistent with lower expression of T cell exhaustion markers (*Figure 6G*). In blood samples, *BRCA1* mutations were associated with enhanced TCR and its downstream signaling, while *BRCA2* mutations were linked to enhanced T and NK-mediated cytotoxicity and NK cell activation receptors (*Figure 6E*).

Furthermore, we performed scTCR-seq to analyze the TCR clonal expansion. Teff, CD8⁺ Trm, and CD4⁺ Trm showed the highest hyperexpanded clonal types (*Figure 6H-I*, *Figure 6—figure supplement 4B-C*), suggesting antigen-driven activation. *BRCA1*-mutant samples had a higher proportion of expanded TCR clonotypes compared to *BRCA2*, across sample types (*Figure 6J*, *Figure 6—figure supplement 4D*). In addition, *BRCA1* mutations were associated with higher TCR diversity in tumor and blood, while *BRCA2* mutations increased TCR richness in blood (*Figure 6—figure supplement 4E*). In *BRCA1*-mutant patient, shared clonotypes tended to expand in paracancerous tissue and were more shared between blood and tumor tissues (*Figure 6K*, *Figure 6—figure supplement 4F-G*). However, in *BRCA2*-mutant patient, shared clonotypes tended to expand in blood and paracancerous tissue (*Figure 6K*, *Figure 6—figure supplement 4F*). *BRCA1* mutations were associated with the expansion of Teff, CD8⁺ Trm, and Tem in solid tissues, while CD4⁺ Trm expansion was more prominent

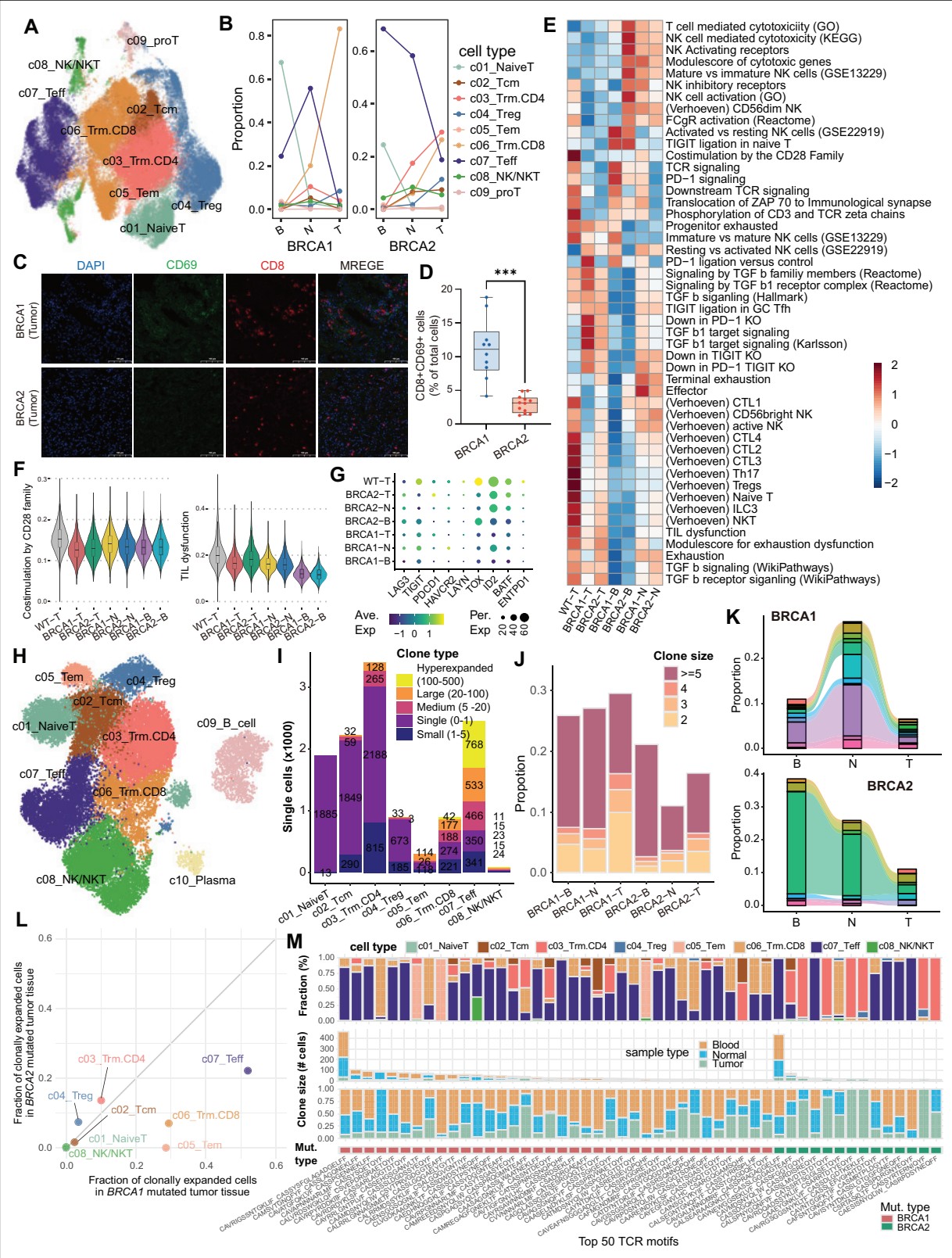

**Figure 6.** T lymphocyte infiltration and TCR clonal expansion analysis. (**A–B**) UMAP visualization (**A**) and proportion (**B**) of T lymphocyte subsets in combined dataset. (**C**) Multiplex immunofluorescence staining for CD8 (red), CD69 (green), and DAPI (blue) on lung tumor sections from patients with *BRCA1* and *BRCA2* mutations. Scale bars, 100 µm. (**D**) Quantification of the relative intensity of the CD8[+] CD69[+] cells. The p value was computed with the two-sided Welch's t test (ns, p>0.05; *, p<0.05; **, p<0.01; ***, p<0.001). (**E–F**) The activity of tissue-specific T cell signatures from Judith Wienke et

*Figure 6 continued on next page*

*Figure 6 continued*

al. (**G**) Terminal exhaustion marker expression levels. (**H**) UMAP of lymphocyte subsets in in-house dataset. (**I**) Quantification of clonal size across sample types. Clonotypes are ranked by expansion level, including: single (1 cell), small (>1 and<5 cells), medium (>5 and<20 cells), large (>20 and<100 cells), and hyperexpanded (>100 and <500 cells). (**J**) Expanded clonotypes distribution in different samples. (**K**) Proportional and dynamic changes in shared clonotypes (top 5 per group) between different samples within the same patient. The colors represent different clonotypes. (**L**) The proportion of clonally expanded cells (≥5 clone size) in *BRCA1/2* mutated tumor tissues for specific cell types. (**M**) The sample and cell proportions of the 50 most abundant TCR motifs.

The online version of this article includes the following figure supplement(s) for figure 6:

**Figure supplement 1.** T lymphocyte activity analysis.

**Figure supplement 2.** Analysis and validation of tissue-resident memory T cells.

**Figure supplement 3.** The distribution of signature score in *BRCA1/2* mutations.

**Figure supplement 4.** Analysis of TCR clonal expansion in T lymphocytes.

---

in *BRCA2* mutations (*Figure 6L*, *Figure 6—figure supplement 4H*). This was especially evident among the top 50 expanded TCR clonotypes (*Figure 6M*). Teff was the most expanded subset in blood, regardless of mutation type (*Figure 6M*, *Figure 6—figure supplement 4I*). Together, these results suggest that *BRCA1* mutations are associated with increased antigen exposure and CD8[+] Trm expansion, while *BRCA2* mutations are linked to CD4[+] Trm expansion in tumor tissue and enhanced T/NK toxicity in blood.

## Targeting a *BRCA1* mutation-associated transcriptional program inhibits LUAD tumor growth in vitro

Our analyses suggested that *BRCA1/2* mutations are associated with distinct transcriptional programs. Then, we identified BRCA mutation-related prognostic factors using Cox regression analysis. The results showed that the factors associated with *BRCA2* mutation were linked to longer patient survival (HR <1, *Supplementary file 1I*), and a *BRCA2* signature score predicted better survival (p=0.0015, *Figure 7—figure supplement 1A*), consistent with upregulation of MHC complex assembly genes. After receiving ICB treatment, lung cancer patients with high *BRCA2* signature scores show longer survival and better responses (p<0.0001; *Figure 7—figure supplement 1B and C* and *Figure 7—figure supplement 2A*). In contrast, a *BRCA1* mutation prognostic factor was associated with shorter survival (HR >1, *Supplementary file 1I*), and its signature score predicted worse survival in patients with LUAD (p=0.0012; *Figure 7A*, *Figure 7—figure supplement 2B*) and NSCLC (p=0.0013; *Figure 7B*), suggesting that *BRCA1* mutation activates a tumor-promoting program. Based on this, we identified four *BRCA1* mutation-associated risk genes (*S100A10*, *LDHA*, *MYL12A*, and *GAPDH*; termed the 4 R genes) for subsequent analysis, which were associated with worse overall survival (*Figure 7—figure supplement 1D and E*). Immunofluorescence experiments on patient tissue sections revealed that *S100A10* was upregulated in *BRCA1*-mutated tumor tissue relative to adjacent non-cancerous tissue (*Figure 7—figure supplement 1D–E*). Previous studies have shown that *S100A10* can promote cancer metastasis by recruiting MDSC cells, and increased *LDHA* activity contributes to tumor immune escape (*Li et al., 2024*; *Certo et al., 2021*). In our study, knockdown of *S100A10*, *LDHA*, and *GAPDH* reduced LUAD cell proliferation in vitro (*Figure 7D–E*).

We next asked whether small molecules could reduce the expression of these risk genes and inhibit LUAD growth. Using gene expression profiles of A549 cells before and after drug treatment from LINCS data resource, we performed cMap analysis to identify drugs that perturb 4 R gene expression. Interestingly, histone deacetylase (HDAC) inhibitors consistently downregulated 4 R genes in A549 cells (*Figure 7F* and *Supplementary file 1J*). Cell growth decreased significantly with increasing HDAC inhibitor concentrations (*Figure 7G and H*, *Figure 7—figure supplement 1G and H*, and *Supplementary file 1K*). Vorinostat (FDA-approved for cutaneous T cell lymphoma) significantly inhibited 4 R gene expression (NCS<-1, p<0.01; *Figure 7F* and *Supplementary file 1J*). Furthermore, vorinostat and belinostat reduced expression of *LDHA*, *S100A10*, and *GAPDH* in A549 cells in a dose-dependent manner (*Figure 7I and J*, *Figure 7—figure supplement 1I*). Other HDAC inhibitors (mocetinostat, entinostat, and SB-939) also showed similar effects on the perturbation of 4 R genes (*Figure 7F–H* and *Supplementary file 1J*). These findings suggested that HDAC inhibitors may target

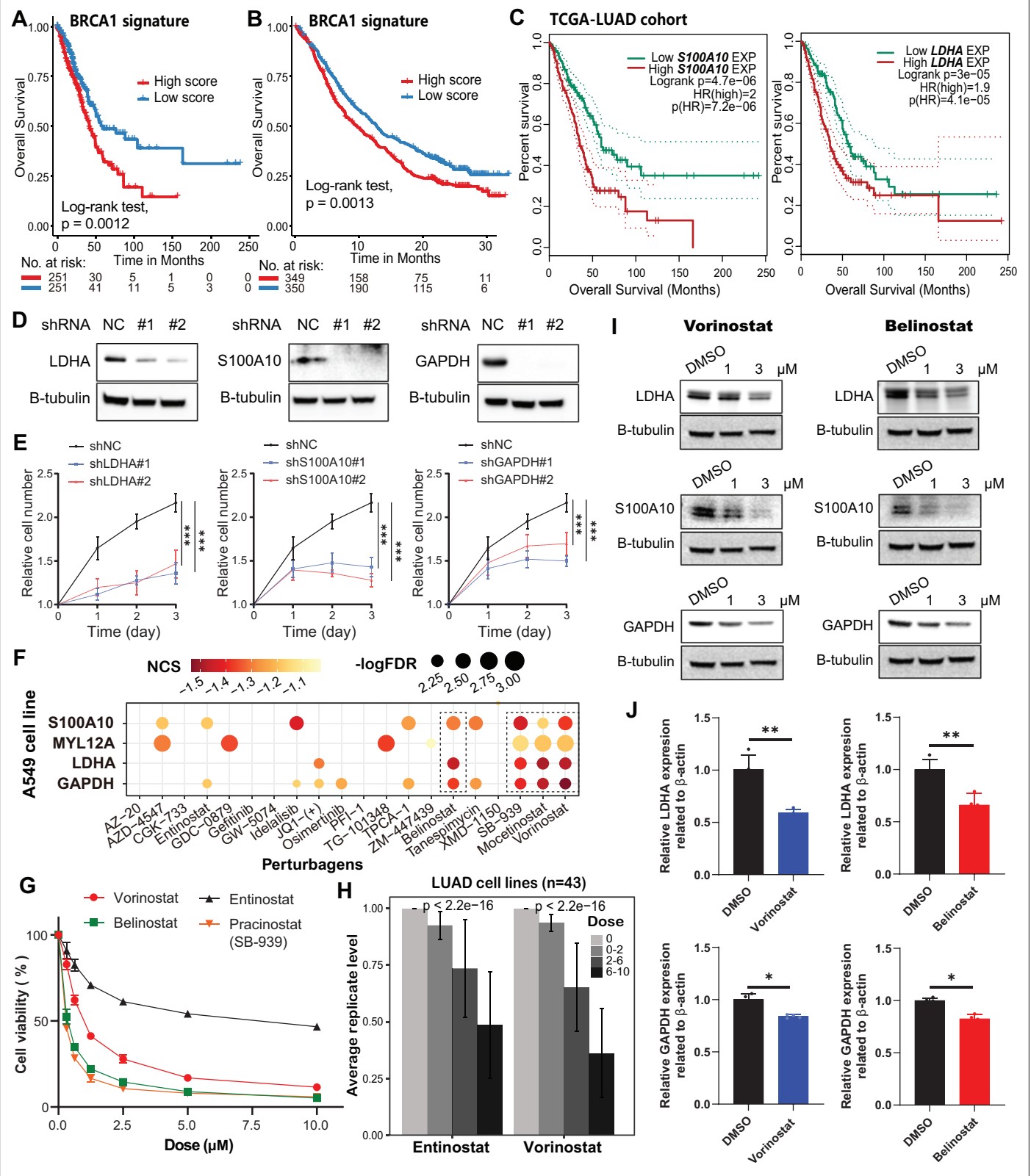

**Figure 7.** Targeting *BRCA1* mutation-related risk genes inhibits tumor growth. (**A–B**) Survival analysis of *BRCA1*-mutant signature score in TCGA-LUAD (**A**) and ICB treated (**B**) cohorts. (**C**) Survival analysis of *S100A10* (left) and *LDHA* (right) in TCGA-LUAD cohort according to the expression median. (**D**) The effects of sh-*LDHA, S100A10,* and *GAPDH* on protein levels in A549 cell line. (**E**) The effects of sh-*LDHA, S100A10, GAPDH,* and vector on cell proliferation were determined by a cell proliferation assay in A549 LUAD cell line. (**F**) The molecular perturbations of 4 R genes in A549 cell line at 10 µM

*Figure 7 continued on next page*

*Figure 7 continued*

concentration (24 hr) from the LINCS data resource. NCS: Normalized connectivity score. (**G**) The effects of HDAC inhibitors concentration on cell proliferation in A549 cell line. (**H**) Cell replication levels after treatment with different concentrations of entinostat (left) and vorinostat (right) in LUAD cell lines. Error bars, means ± SEM. Significance level was calculated by Kruskal-Wallis test. (**I**) The protein levels of *LDHA*, *S100A10*, and *GAPDH* after treatment with vorinostat (left) and belinostat (right) in A549 cell line were measured using western blotting. (**J**) The mRNA levels of *LDHA* and *GAPDH* after treatment (3 μM) with vorinostat (left) and belinostat (right) in A549 cell line were measured using qPCR (n=4 per group). Error bars, means ± SEM. Significance levels were determined by Student's t-test. For all statistical tests, *p≤0.05; **p≤0.01; ***p≤0.001.

The online version of this article includes the following source data and figure supplement(s) for figure 7:

**Source data 1.** PDF file containing original western blots for *Figure 7D and I*.

**Source data 2.** Original files for western blot analysis displayed in *Figure 7D and I*.

**Figure supplement 1.** Characteristic analysis of *BRCA1/2* mutation-related signatures.

**Figure supplement 2.** Multivariable Cox regression of the *BRCA1/2* signature score.

*BRCA1* mutation-associated risk genes and inhibit LUAD tumor growth in vitro, offering a preliminary therapeutic hypothesis for further testing.

## Discussion

Our study provides a comprehensive molecular and cellular characterization of the TME in LUAD with *BRCA1* and *BRCA2* mutations, using large-scale multi-omics data and single-cell data. We describe transcriptional and immune patterns associated with these mutations, which may contribute to genomic instability, immune cell infiltration, and immune modulation within the TME. These observations offer hypotheses for understanding HRR deficiency in LUAD and for developing potential therapeutic strategies, including ICB and targeted therapies. HRR deficiency is a key contributor to tumorigenesis and treatment response in multiple cancers (*Hoppe et al., 2018*). Although HRR deficiency has been correlated with enhanced sensitivity to DNA-damaging agents and ICB in cancers such as breast, ovarian, and NSCLC, our study suggests that in LUAD, *BRCA1/2* mutations may be associated with distinct TME features that could influence immunotherapy effectiveness. This underscores the complexity of BRCA-driven biology in LUAD.

A key aspect of this study is the single-cell transcriptomic profiling of *BRCA1/2*-mutant LUAD tumors, paracancerous tissues, and peripheral blood patients. This enabled high-resolution examination of immune signatures, though the sample size limits generalizability. *BRCA1* mutations were associated with the upregulation of cGAS-STING and type I IFN response genes, which can enhance innate immune responses, antigen presentation, and cytotoxic T-cell activity (*Hu et al., 2024*). *BRCA1*-mutant tumors showed higher CD8$^+$ T-cell activation and MHC-I expression, possibly due to increased neoantigen load. This suggests that *BRCA1*-mutant tumors may be more immunogenic and potentially responsive to immunotherapy. In contrast, *BRCA2* mutation was associated with upregulated MHC-II expression, which plays a role in antigen presentation and CD4$^+$ T-cell recruitment (*Axelrod et al., 2019*), suggesting a shift toward adaptive immunity and antigen-presenting cell involvement. *BRCA2*-mutant tumors showed higher CD4$^+$ T-cell activation, particularly inflammatory subsets, and were associated with increased NK cell activity and T-cell cytotoxicity in blood, suggesting systemic immune engagement.

Despite these immune signatures, *BRCA1/2* mutations were associated with downregulation of CD28 family co-stimulatory molecules, essential for effective T-cell activation (*Burke et al., 2024*). This may indicate adaptive resistance mechanisms. ICB may overcome such resistance (e.g. blocking PD-L1), unleashing the pre-existing, infiltration-potentiated immune response (*Figure 1G*). We identified two Trm subsets (CD8$^+$ Trm and CD4$^+$ Trm) as potential biomarkers for ICB response in NSCLC. *BRCA1* mutation was associated with CD8$^+$ Trm TCR expansion, while *BRCA2* mutation was linked to CD4$^+$ Trm expansion, which may contribute to maintaining a chronic inflammatory TME (*Chi et al., 2024*; *Kruse et al., 2023*). Given the observed T-cell infiltration and IFN-driven immune activation, *BRCA1*-mutant tumors might initially respond well to anti-PD-1/PD-L1 therapies. However, *BRCA1* mutations also activated risk genes (*S100A10*, *LDHA*, *GAPDH*, and *MYL12A*). We identified HDAC inhibitors as potential agents targeting this program. HDAC inhibitors can modulate chromatin accessibility, reduce immune suppression, and enhance tumor immunogenicity, suggesting that combining

them with ICB might enhance immune response and overcome resistance mechanisms in *BRCA1*-driven tumors.

Despite the comprehensive multi-omics and single-cell analyses conducted in this study, several limitations should be acknowledged. Although we included both tumor tissues and matched paracancerous and blood samples, the sample size remains modest, which may limit the statistical power and generalizability of our findings. Therefore, our results should be interpreted as preliminary, and further studies with larger, independent cohorts are required to validate these observations. Single-cell RNA-seq and TCR-seq analyses in this study provide high-resolution insights into the cellular and clonal dynamics of the TME; the functional validation of key mechanisms remains largely correlative. While our in vitro experiments provide valuable mechanistic insight, the lack of in vivo validation cannot fully recapitulate the complex TME. Future studies utilizing murine models or patient-derived organoids are essential to establish causal relationships and elucidate the underlying molecular pathways. The efficacy and safety of combining HDAC inhibitors with ICB in *BRCA1*-mutant LUAD patients remain unexplored and represent an important avenue for future research.

Overall, this study provides a high-resolution molecular map of the TME in *BRCA1/2*-mutant LUAD, highlighting distinct immune and transcriptional signatures associated with each mutation in the profiled cases. By uncovering these molecular distinctions, our study offers a hypothesis-generating framework for understanding HRR deficiency in LUAD and lays the groundwork for future validation and precision immunotherapy strategies tailored to *BRCA1/2* mutation status.

# Materials and methods

## Human specimens

Patient-derived samples for this study were collected at Guangzhou Institute of Cancer Research, the Affiliated Cancer Hospital, Guangzhou Medical University, Guangdong, China. All patients were treatment-naïve and had invasive lung adenocarcinoma and PD-L1 negative confirmed based on immunohistochemistry. Fresh tumor and adjacent normal tissues of patients were collected through surgery. This study was approved by the Ethics Committee of Affiliated Cancer Hospital and Institute of Guangzhou Medical University (Approval No. KY-2025003) in compliance with the international guidelines. All patients have signed informed consent. The samples were divided into two parts, one for single-cell sequencing and the other for whole exome sequencing. In addition, we collected and obtained public datasets of multiple cohorts. The genomic variation, gene expression, and clinical data of LUAD patients (n=564) of the cancer genome atlas (TCGA) program were downloaded from cBioPortal data resource (https://www.cbioportal.org/). The genomic variation and survival data of LUAD in individuals of East Asian ancestry (OncoSG cohort, n=305; *Chen et al., 2020*) also from cBioPortal. The mutation and survival data of LUAD patients receiving ICB treatment from SU2C MARK cohort (n=309; *Ravi et al., 2023*). The gene expression, survival, and response data of NSCLC patients receiving ICB treatment from OAK cohort (n=699; *Patil et al., 2022*). TMB, HRD score, and neoantigen load data for TCGA-LUAD patients were obtained from a previous study (*Thorsson et al., 2018*). The HRD score was determined by summing specific genomic alterations, including loss of heterozygosity (LOH), large-scale state transitions (LST), and telomeric allelic imbalances (TAI). TMB was defined as the total number of somatic nonsynonymous mutations per megabase of the exome captured by the sequencing panel. Neoantigen load was predicted by NetMHCpan using the patient's HLA typing and the identified somatic mutations.

## DNA extraction and library construction of whole exome sequencing

Genomic DNA was extracted from lung adenocarcinoma tissues and blood using CretMag Multi Sample DNA Kit. Genomic DNA was captured using Agilent SureSelect Human All Exon v6 library following the manufacturer's protocol (Agilent Technologies, USA). DNA libraries were constructed following the protocols provided by Illumina. Then these libraries were sequenced on the Illumina sequencing platform (Illumina NovaSeq X plus), and 150 bp paired-end reads were generated. The whole exome sequencing was conducted by OE Biotech Co., Ltd. (Shanghai, China).

## Somatic variant identification

The raw reads (fastq) of whole exome sequencing were pre-processed and trimmed with fastp (Version: 0.23.4) based on default parameters. Clean reads were aligned to the reference human genome (GRCh38) utilizing the BWA (version 0.7.17). The mapped reads were sorted and indexed by using SAMtools (Version 1.5). The GATK (Version 4.0.5.1) was used for recalibration of the base quality score and for single nucleotide polymorphism (SNP) and insertion/deletion (INDEL) realignment, and used for marking duplicate reads, to obtain analysis-ready BAM files. The final BAM files were used as input files for variant calling. Identification of somatic mutations using the GATK Mutect2 method was based on matched blood sample and population germline resource from gnomAD (https://gnomad.broadinstitute.org/). ANNOVAR was used to annotate variations during SNP & INDEL calling (*Wang et al., 2010*).

## Single-cell suspension preparation

The solid tissues (including tumor and adjacent tissues) were processed to prepare single-cell suspensions. The tissue samples were cut into small pieces under aseptic conditions, and the cells were then washed twice with pre-cooled RPMI 1640 medium supplemented with 0.04% BSA. The tissues were cut into small pieces of approximately 0.5 mm$^3$ using surgical scissors, placed in freshly prepared hydrolase, and digested at a constant temperature of 37 °C for 60 min, mixing every 10 minutes. The tissue was filtered two times using a cell sieve and centrifuged (300 × $g$) for 5 min at 4 °C. After resuspending the sediment with an appropriate amount of medium, an equal volume of erythrocyte lysis buffer (MACS, catalog number: 130-094-183) was added. After mixing, the cells were incubated at 4 °C for 10 min and centrifuged for 5 min (300 × $g$), and the number of viable cells, as determined by Trypan blue staining, was determined with a hemocytometer. Human fresh peripheral blood was collected in an anticoagulant tube and diluted with PBS at 1:1. Then, the sample was added to the Ficoll separation medium (Merck) and centrifuged for 20 min (75 × $g$). After centrifugation, a white-grayish layer consisting of mononuclear cells was found on top of the Ficoll-Paque. We carefully collected the mononuclear cells with a Pasteur pipette, transferred them to a new centrifuge tube, and washed them by centrifuging for 10 min (300 × $g$). The supernatant was discarded and resuspended in RPMI-1640 supplemented with 0.04% BSA. The number of viable cells, as determined by Trypan blue staining, was determined with a hemocytometer.

## Single-cell RNA and V(D)J sequencing

The cell concentration of the freshly prepared single-cell suspension was adjusted to 700–1200 cells/µl. The cDNA library amplification was performed according to the operating instructions of 10×Genomics Chromium Next GEM Single Cell 5' Reagent Kits v2.0 (PN-1000263). DNA library construction was performed using the 10 x Genomics Chromium Single Cell 5' Kit (PN-1000190) following the manufacturer's guidelines. Purified cDNA libraries were sequenced using the paired-end 150 mode on an Illumina platform (Illumina NovaSeq X plus). For T-cell receptor (TCR), T cell V(D)J enrichment was performed using the 10×Genomics single-cell V(D)J Enrichment Kit (TCR, PN-1000005) according to the human T cell operating instructions. The TCR library was amplified using the Chromium Single Cell TCR Amplification Kit (TCR, PN-1000252), and the experimental operation was performed according to the product instructions. The constructed libraries were subjected to high-throughput sequencing (Illumina NovaSeq X plus) using the paired-end 150 mode at OE Biotech Co. Ltd (Shanghai, China).

## scRNA-seq and scTCR-seq data processing

Raw read files were processed with Cell Ranger 7.1.0 and mapped to the GRCh38 genome assembly and counted with GRCh38 annotations. Unique molecular identifiers (UMIs) were calculated by using the 'cellranger count' function. As a result, a cellular gene expression matrix containing the number of gene UMIs detected in each cell was generated. In addition, we took several steps to obtain high-quality data. First, cells with high mitochondrial gene expression were removed, as dead cells often exhibit mitochondrial contamination (*Ilicic et al., 2016*). Specifically, we followed the previous method and used the median-centered median absolute deviation (MAD)-variance normal distribution to fit the expression levels of mitochondrial genes and removed cells with expression levels significantly higher than expected (Benjamini-Hochberg-corrected FDR <0.01; *Pijuan-Sala et al., 2019*). Second, we removed cells with less than 500 or more than 6000 detected genes. Third, we used DoubletFinder

(*McGinnis et al., 2019*) and scDblFinder (*Germain et al., 2021*) methods to identify and remove potential doublets. The 90th percentile of the doublet score of DoubletFinder was used as the cutoff value. The expected doublet rate is 10% in the scDblFinder method. Raw TCR reads were aligned to the GRCh38 reference genome, and consensus TCR annotation was performed using the 'cellranger vdj' function (Cell Ranger 7.1.0). TCR annotation was performed using the default parameters of the 10 X cellranger vdj pipeline. The R package scRepertoire was employed for clonotype assignment and dynamics analysis.

## Normalization, clustering, and visualization of scRNA-seq data

The gene expression matrix of the selected high-quality cells was subjected to downstream analysis using the R package Seurat (v5.0.3; *Hao et al., 2024*). The main steps include the following. First, only genes expressed in more than five cells were retained. Second, the raw UMI count matrix was log-normalized and scaled to 10,000 using the 'LogNormalize' function. Third, 2000 highly variable genes were identified using the 'FindVariableFeatures' function based on the normalized matrix. Fourth, the gene expression matrix was scaled and centered using the 'ScaleData' function. Fifth, unsupervised clustering was performed by constructing a shared nearest neighbor (SNN) graph using the 'FindNeighbors' function and the Louvain algorithm. The first 30 principal components were considered, and the clustering resolution was 1.6. For uniform manifold approximation and projection (UMAP) visualization, the first two dimensions of UMAP were calculated using the 'RunUMAP' function. Sixth, differential expression analysis between clusters was performed using the 'FindAllMarkers' function from Seurat (using default parameters but setting min.pct to 0.5). Differential expression genes for specific clusters were identified using the 'FindMarkers' function, with a threshold of $|avg\_log2FC|\geq0.5$ and adjusted p-value $\leq0.01$. Finally, two complementary approaches are used to annotate cell clusters: (1) canonical markers; (2) markers that were in the top rank of differentially expressed genes.

## Signature score calculation at single-cell and bulk level

Signature (module) score at single-cell level was calculated for selected gene-sets (including pathways and signatures) using the 'AddModuleScore_UCell' function from the UCell R package. The pathways or signatures involved in this study include: immune pathways and receptors from ImmPort database (https://immport.org/shared/home), tissue-specific T cell signatures (*Wienke et al., 2024*), and known cancer meta-programs (*Gavish et al., 2023*) from previous studies, as well as several specific signatures from the molecular signatures database (MSigDB) human collections. In addition, tumor tissue upregulated and downregulated gene signatures are the top 60 up-regulated genes and the bottom 60 down-regulated genes, respectively, calculated using the TCGA-LUAD cohort. CD8$^+$ Trm and CD4$^+$ Trm signatures were identified based on 'FindMarkers' function at single-cell level. Signature score at bulk level was calculated using single sample gene set enrichment analysis (ssGSEA) algorithm in R package GSVA.

## Functional enrichment analysis

Differential expression analysis between malignant cells and normal cells was performed, and the up-regulated genes of malignant cells were screened according to the thresholds $avg\_log2FC \geq0.5$ and Padj $\leq0.01$. According to the descending sorting of avg_log2FC, the top 200 genes of malignant cells were selected. The GO BP functional enrichment analysis was performed using the R package clusterProfiler (the thresholds used were $p\leq0.001$ and Padj $\leq0.01$). Padj calculations were based on Benjamini-Hochberg correction. The same approach was used for the comparison between the two types of malignant cells and the comparison between malignant cells in *BRCA1* and *BRCA2* mutation samples. In addition, we obtained immune signatures and gene modules of immune cell type from MSigDB hallmarks, ImmPort database, and previous studies (*Mariathasan et al., 2018*; *Bindea et al., 2013*). Gene set enrichment analysis (GESA) was employed to compare the immune response profile between *BRCA1* and *BRCA2* mutations using the R package enrichplot.

For all malignant cells, we performed pathway enrichment analysis using ActivePathways (v.2.0.3) R package for the terms from MSigDB (human collection C5) based on the gene sets of two meta-programs. The terms were removed once the number of genes was less than 10 or more than 500. Terms with BH-corrected significance adjusted $p\leq0.05$ are considered to be significantly enriched by

the gene sets of two meta-programs and will be retained. The significantly enrichment results were used as input to the EnrichmentMap plugin in cytoscape (v.3.9.1) software to draw a network diagram.

## Single-cell copy number variation inference

The inferCNV (v.1.19.1) R package was used to distinguish malignant cells by inferring chromosomal CNVs based on the single-cell expression data. The cells from CD8[+] T cells as normal reference were used to estimate CNVs for the population of malignant cells. We prepared gene order files containing the chromosomal start and end positions of each gene from the GRCh38 assembly as input to the 'gene_order_file' parameter in the 'CreateInfercnvObject' function. To perform the infercnv operation, the count matrix and annotation file are input to create the infercnv object, and then inferCNV is run with cutoff = 0.1 in the 'run' function.

## Identification of potential drugs targeting BRCA1-mutant genes

This study identified potential drug small molecules based on *BRCA1* mutation-related cancer-promoting factors. Gene expression profiles of lung adenocarcinoma cell lines before and after small molecule drug treatment were obtained from the NIH library of integrated network-based cellular signatures (LINCS) data resource. The connectivity Map (cMap) analysis was employed to identify potential drugs associated with molecular perturbations of four *BRCA1* mutation prognostic risk factors (4 R genes; *Liao et al., 2022*). To demonstrate the inhibitory effect of HDAC inhibitors on the growth of lung adenocarcinoma cell lines, the replication levels at different HDAC inhibitor concentrations were obtained from the cancer cell line encyclopedia (CCLE) data resource. However, due to limited data, we only obtained data on two HDAC inhibitors, entinostat and vorinostat, in 43 lung adenocarcinoma cell lines. Since drug concentrations (C) were divergent, we categorized them into 4 groups: 0 µM$<$C $\leq$ 2 µM ([0–2]), 2 µM$<$C $\leq$ 6 µM ([2-6]), 6 µM$<$C $\leq$ 10 µM ([6-10]) and vehicle control (C=0 µM). The mean of each group is the average replication level of the cell line.

## Cell culture and reagents

The human lung cancer cell line A549 (#SCSP-503) and the human embryonic kidney cell line HEK293T (#SCSP-502) were purchased from the Type Culture Collection of the Chinese Academy of Sciences, China. Short Tandem Repeat (STR) analyses were performed to authenticate the identity of each cell line used in this study. The A549 cells were cultured in RPMI-1640 medium (#C11875500BT; Gibco). The HEK293T cells were cultured in Dulbecco's Modified Eagle's medium (#C11995500BT; Gibco). Both cell lines were supplemented with 10% fetal bovine serum (#S712-012S; Lonsera), and grown in a humidified 5% $CO_2$ atmosphere at 37 °C. All cell lines utilized in this study were confirmed to be free of contamination.

## Vectors and lentiviral transfection

All the short hairpin RNAs (shRNA) were cloned into the pLKO.1 vector. The target sequences are as follows:

> LDHA#1: GCCACAGATTTACCCGTGGAT,
> LDHA#2: GCCAACAACTTGTGTCTCAAT,
> S100A10#1: TAAGGAGCCAAATACCTTGCG,
> S100A10#2: CATGAAACACAAACGGCAAAT,
> GAPDH#1: GTGCGGAGTGTAATCAGTATT,
> GAPDH#2: TCAGGTTGTACGGGATCAAAT.

Lentiviral particles were produced by co-transfecting HEK293T cells with the shRNA plasmid and the packaging vectors psPAX2 and pMD2.G at a ratio of 4:3:1, respectively, using JetPRIME In Vitro DNA Transfection Reagent (#101000046 (114-15); Polyplus). The culture media was changed after 6 hr. At 48 hr post-medium change, the virus-containing medium was collected and concentrated. Viral supernatants were centrifuged at 1500 × *g* for 45 min, and the viral pellets were resuspended in DMEM. The lentivirus was stored at –80 °C until further use. The knockdown efficiency of the lentiviral shRNA clones was determined by western blotting.

## Lentiviral infection

$1\times10^5$ A549 cells were seeded into each well of a six-well plate and simultaneously treated with 400 µL of lentivirus and 8 µg/mL polybrene (#28728-55-4; Selleck). The plates were incubated at 37 °C for

12 hr. The cells were then passaged and seeded into new culture dishes. At 24 hr post-passage, the cells infected with the lentivirus were selected with 1 μg/mL puromycin (#P8833-10mg; Sigma-Aldrich) for 3 days.

## Cell proliferation assay

Cell proliferation was measured over a 4-day time course using the CellTiter-Glo Luminescent Cell Viability Assay kit (#G9242; Promega). For this assay, 1000 cells were seeded into each well of a 96-well plate containing 100 μL of medium per well and cultured under normal conditions. At each time point, plates were cooled to room temperature, and then 100 μL of CellTiter-Glo reagent was added to each well. The contents were mixed on an orbital shaker for 2 min to induce cell lysis. The plate was then incubated at room temperature for 10 min to stabilize the luminescent signal. Finally, luminescence was measured using a microplate reader (#SLXFA-SN; Gene Company). All data were normalized to Day 1 and are presented as mean ± SD.

## Drug treatment

For the cell viability assay, 5000 A549 cells were seeded into 96-well plates containing 100 μL of medium per well. After 24 h of culture at 37 °C in a 5% $CO_2$ atmosphere, the cells were treated with gradient concentration of the corresponding agents (Vorinostat [S1047], Belinostat [S1085], Entinostat [S1053], Pracinostat [S1515]; all from Selleck Chemicals) for 48 hr. Cell viability was determined using the CellTiter-Glo Luminescent Cell Viability Assay kit (#G9242; Promega) according to the manufacturer's instructions. To detect the protein levels of the targeted genes induced by drug treatment, $4{\times}10^5$ A549 cells were seeded into each 60 mm cell culture dish. After 24 hr of culture at 37 °C in a 5% $CO_2$ atmosphere, the cells were treated with 1 μM and 3 μM Vorinostat or Belinostat for 2 days. The cells were then harvested to assess the protein levels of the targeted genes.

## Western blotting

To assess the efficiency of lentiviral shRNA clones and the effects of drug treatments, cells were washed twice with PBS, harvested by trypsinization, and collected by centrifugation at 1500 × $g$ for 5 min. The cell pellets were washed twice with PBS, resuspended in RIPA buffer (#P0013B; Beyotime) supplemented with a protease inhibitor cocktail (#78440; Invitrogen), incubated on ice, and then boiled for 10 min at 100 °C. Protein lysates were resolved on 4–12% YoungPAGE Precast Gels (#PK0931; GenScript) and transferred onto a 0.45 μm Immobilon-P PVDF membrane (#IPVH00010; Merck Millipore). The membranes were blocked for 1 hr in TBST (#BL602A; Biosharp) containing 5% non-fat dry milk and subsequently incubated with primary antibodies for 1 hr at room temperature or overnight at 4 °C. Following incubation with appropriate horseradish peroxidase (HRP)-conjugated secondary antibodies, signals were detected using an enhanced chemiluminescence (ECL) reagent (#CW0049M; CWBIO). Images were captured using a MiniChemi580+Imaging System (SINSAGE). The following antibodies were used: GAPDH Rabbit mAb (#A19056, 1:100,000; Abclonal), S100A10 Rabbit mAb (#A13634, 1:1000; Abclonal), LDHA Rabbit mAb (#A0861, 1:1000; Abclonal), β-Tubulin Rabbit mAb (#A12289, 1:10000; Abclonal), and HRP-conjugated Goat Anti-Rabbit IgG (H+L) (#111-035-003, 1:10000; Jackson).

## RNA isolation and RT-qPCR

Total RNA was isolated from cells using TRIzol Reagent (#15596018; Thermo Fisher Scientific). RNA concentration was measured with a NanoDrop 2000 spectrophotometer (Thermo Fisher Scientific, USA). Total RNA was reverse transcribed into cDNA using the HiScript III 1st Strand cDNA Synthesis Kit with gDNA wiper (#R312-02; Vazyme), following the manufacturer's instructions. Quantitative reverse transcription PCR (qRT-PCR) was performed on the SLAN-96P Real-Time PCR System using ChamQ SYBR qPCR Master Mix (#Q311-02; Vazyme) and gene-specific primers according to the manufacturer's protocol. The primer sequences used for qPCR are as follows:

> LDHA-fwd: 5'-ACCGTGTTATTGGAAGCGGT –3',
> LDHA-rev: 5'-CTCCATGTTCCCCAAGGACC –3',
> GAPDH-fwd: 5'-GTCAAGGCTGAGAACGGGAA –3',
> GAPDH-rev: 5'-AAATGAGCCCCAGCCTTCTC –3',
> β-actin-fwd: 5'-CACCATTGGCAATGAGCGGTTC –3',

β-actin-rev: 5'-AGGTCTTTGCGGATGTCCACGT –3'.

## Statistics and survival analysis

All statistical analyses were conducted using R software (version 4.2.3, http://www.r-project.org). The two-sided Wilcoxon rank-sum test was employed to count and compare the differences between the two groups. Kruskal-Wallis test was used to compare the differences among the three groups. The significance level of discrete variables was calculated by Chi-squared test. The identification of prognostic factors associated with *BRCA1* and *BRCA2* mutations was based on the up-regulated genes of the corresponding mutations using univariate Cox regression analysis. Survival analysis was performed by the log-rank test and compared using Kaplan-Meier curves. Statistical significance for pairwise comparisons across the multiple groups was assessed using the Benjamini-Hochberg false discovery rate (FDR) correction. For all statistical tests, $*p \leq 0.05$; $**p \leq 0.01$; $***p \leq 0.001$; $****p \leq 0.0001$.

## Acknowledgements

The authors wish to thank the staff members from Affiliated Cancer Hospital and Institute of Guangzhou Medical University. High-throughput sequencing was performed by the OE Biotech Co., Ltd. (Shanghai, China). The contribution of the bioinformatics core facility at Guangzhou Medical University is gratefully acknowledged. This study was supported by the National Natural Science Foundation of China (Grant No. 82403816), the China Postdoctoral Foundation (Grant No. 2023M740846), the Guangdong Basic and Applied Basic Research Foundation (Grant No. 2023A1515110297), the Beijing Vlove Charity Foundation (Grant No. RXYS2025-0100630114), and the Fundamental Research Funds for the Provincial Universities (Grant No. 2023-KYYWF-0212).

## Additional information

### Funding

| Funder | Grant reference number | Author |
| --- | --- | --- |
| National Natural Science Foundation of China | 82403816 | Gaoming Liao |
| China Postdoctoral Science Foundation | 2023M740846 | Gaoming Liao |
| Fundamental Research Funds for the Provincial Universities | 2023-KYYWF-0212 | Xionghai Qin |
| Beijing Vlove Charity Foundation | RXYS2025-0100630114 | Gang Xu |
| Guangdong Basic and Applied Basic Research Foundation | 2023A1515110297 | Gaoming Liao |

The funders had no role in study design, data collection and interpretation, or the decision to submit the work for publication.

### Author contributions

Gaoming Liao, Conceptualization, Data curation, Software, Formal analysis, Funding acquisition, Visualization, Methodology, Writing – original draft; Xinbin Yang, Investigation, Methodology, Writing – review and editing; Qi Liu, Validation, Investigation, Writing – review and editing; Shufeng Nan, Software, Validation; Yan Liu, Software; Jinwei Li, Si Huang, Wang Ning, Validation; Xionghai Qin, Conceptualization, Supervision, Funding acquisition, Writing – review and editing; Gang Xu, Conceptualization, Supervision, Project administration, Writing – review and editing

### Author ORCIDs

Gaoming Liao ⬡ https://orcid.org/0009-0005-6414-3670

## Ethics

This study was approved by the Ethics Committee of Affiliated Cancer Hospital and Institute of Guang-zhou Medical University (Approval No. KY-2025003) in compliance with the international guidelines. All patients have signed informed consent.

Reviewer #1 (Public review): https://doi.org/10.7554/eLife.110662.3.sa1
Reviewer #2 (Public review): https://doi.org/10.7554/eLife.110662.3.sa2
Author response https://doi.org/10.7554/eLife.110662.3.sa3

# Additional files

## Supplementary files

Supplementary file 1. Supplementary tables containing somatic mutations, differential expressed genes, MP1 and MP2 genes, functional enrichment results, prognostic factors, CMap analysis results, and cell line average replication levels. (**A**) Somatic mutations identified from whole exome sequencing. (**B**) Differentially upregulated genes from malignant cells relative to other epithelial cells. (**C**) Functional enrichment analysis of differentially upregulated genes from malignant cells relative to other epithelial cells. (**D**) Differentially expressed genes from malignant 1 cell type relative to malignant 2 cell type. (**E**) Functional enrichment analysis of differentially expressed genes from malignant 1 cell type relative to malignant 2 cell type. (**F**) MP1 and MP2 genes. (**G**) Function enrichment of MP1 and MP2 genes. (**H**) Differentially expressed genes of *BRCA1/2* mutated malignant cells. (**I**) Prognostic factors associated with *BRCA1* and *BRCA2* mutations. (**J**) CMap analysis of the 4 *BRCA1* mutation-related risk genes in lung adenocarcinoma cell lines. (**K**) Average replication levels of lung adenocarcinoma cell lines at different concentrations of entinostat and vorinostat.

MDAR checklist

## Data availability

The raw data of single-cell sequencing (including scRNA-seq and scTCR-seq) and the processed data generated in this study have been deposited at the Gene Expression Omnibus (GEO) repository, with the accession code GSE292700. The sources of all other datasets in this study are presented in the Materials and Methods section. Open-source R packages and software, as well as standard work-flows were used in this study. No previously unreported custom code was developed for the analyses presented.

The following dataset was generated:

| Author(s) | Year | Dataset title | Dataset URL | Database and Identifier |
|---|---|---|---|---|
| Liao G, Yang X, Liu Q, Nan S, Liu Y, Li J, Huang S, Ning W, Qin X, Xu G | 2026 | Molecular architecture of the tumor microenvironment caused by BRCA1 and BRCA2 somatic mutations in human lung adenocarcinoma | https://www.ncbi.nlm.nih.gov/geo/query/acc.cgi?acc=GSE292700 | NCBI Gene Expression Omnibus, GSE292700 |

The following previously published datasets were used:

| Author(s) | Year | Dataset title | Dataset URL | Database and Identifier |
|---|---|---|---|---|
| Hoadley KA, Yau C, Hinoue T, Wolf DM, Lazar AJ, Drill E, Shen R, Taylor AM, Cherniack AD, Thorsson V, Akbani R, Bowlby R, Wong CK, Wiznerowicz M, Sanchez-Vega F, Robertson AG, Schneider BG, Lawrence MS, Noushmehr H, Malta TM, Laird PW, Cancer Genome Atlas Network, Stuart JM, Benz CC, Laird PW | 2018 | Lung Adenocarcinoma (TCGA, PanCancer Atlas) | https://www.cbioportal.org/study/summary?id=luad_tcga_pan_can_atlas_2018 | cBioPortal, luad_tcga_pan_can_atlas_2018 |
| Chen JB, Yang HC, Lb A | 2020 | Lung Adenocarcinoma (OncoSG, Nat Genet 2020) | https://www.cbioportal.org/study/summary?id=luad_oncosg_2020 | cBioPortal, luad_oncosg_2020 |
| The Broad Institute Lab | 2018 | L1000 Connectivity Map perturbational profiles from Broad Institute LINCS Center for Transcriptomics LINCS PHASE *II* (n=354,123; updated March 30, 2017) | https://www.ncbi.nlm.nih.gov/geo/query/acc.cgi?acc=GSE70138 | NCBI Gene Expression Omnibus, GSE70138 |

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
