## [Editor Report · eLife Assessment]

This **important** study investigates the impact of BRCA1/2 mutations on immunotherapy in lung adenocarcinoma using multi-omics approaches. The detailed genetic analysis of two cancer genes (BRCA1 and BRCA2) demonstrated their new roles in causing the tumor microenvironment in lung cancer. The **solid** findings of this study provide an essential foundation for further developing drugs targeting BRCA1/2 in lung cancer therapy.

---

## [Referee Report · Reviewer #1 (Public review)]

Summary:

Liao et al. performed a large-scale integrative analysis to explore the function of two cancer genes (BRCA1 and BRCA2) in lung cancer, which is one of the cancers with an extremely high mortality rate. The detailed genetic analysis demonstrated new roles of BRCA1/2 in causing the tumor microenvironment in lung cancer. In particular, the discovery of different mechanisms of BRCA1 and BRCA2 provides an essential foundation for developing drugs that target BRCA1 or BRCA2 in lung cancer therapy.

Strengths:

(1) This study leveraged large-scale genomic and transcriptomic datasets to investigate the prognostic implications of BRCA1/2 mutations in LUAD patients (~2,000 samples). The datasets range from genomics to single-cell RNA-seq to scTCR-seq.

(2) In particular, the scTCR-seq offers a powerful approach for understanding T cell diversity, clonal expansion, and antigen-specific immune responses. Leveraging these data, this study found that BRCA1 mutations were associated with CD8+ Trm expansion, whereas BRCA2 mutations were linked to tumor CD4+ Trm expansion and peripheral T/NK cell cytotoxicity.

(3) This study also performed a comprehensive analysis of genomic variation, gene expression, and clinical data from the TCGA program, which provides an independent validation of the findings from LUAD patients newly collected in this study.

(4) This study provides an exemplary integration analysis using both computational biology and wet bench experiments. The experimental testing in the A549 cell line further supports the robustness of the computational analysis.

(5) The findings of this study offer a comprehensive view of the molecular mechanisms underlying BRCA1 and BRCA2 mutations in LUAD. BRCA1 and BRCA2 are two well-known cancer-related genes in multiple cancers. However, their role in shaping the tumor microenvironment, particularly in lung cancer, is largely unknown.

(6) By focusing on PD-L1-negative LUAD patients, this study demonstrated the molecular mechanisms underlying resistance to immune therapy. These new insights highlight new opportunities for personalized therapeutic strategies to BRCA-driven tumors. For example, they found histone deacetylase (HDAC) inhibitors consistently downregulated 4-R genes in A549 cells.

(7) The deposition of raw single-cell sequencing (including scRNA-seq and scTCR-seq) data will provide an essential data resource for further discovery in this field.

Comments on revisions:

The author has revised accordingly. I have no further comments.

---

## [Referee Report · Reviewer #2 (Public review)]

Summary:

This study investigates the impact of BRCA1/2 mutations on immunotherapy in lung adenocarcinoma using multi-omics approaches. The work highlights distinct roles of BRCA1 and BRCA2 mutations in shaping immune-related processes, and is logically structured with clearly presented analyses. However, the conclusions rely primarily on descriptive computational analyses and would benefit from additional immunological validation.

Strengths:

By integrating public datasets with in-house data, this study examines the impact of BRCA1/2 mutations on immunotherapy in lung adenocarcinoma from multiple perspectives using multi-omics approaches. The analyses are diverse in scope, with a clear overall logic and a well-organized structure.

Weaknesses:

The study is largely descriptive and would benefit from additional immunological experiments or validation using in vivo models. The fact that the BRCA1 and BRCA2 samples were each derived from a single patient also limits the robustness of the conclusions.

Comments on revisions:

The authors have addressed my concerns satisfactorily

---

## [Author Response]

The following is the authors’ response to the original reviews.

**eLife Assessment**
This important study investigates the impact of BRCA1/2 mutations on immunotherapy in lung adenocarcinoma using multi-omics approaches. The detailed genetic analysis of two cancer genes (BRCA1 and BRCA2) demonstrated new roles for these genes in causing the tumor microenvironment in lung cancer. Further experimental explorations of the immune-related changes may still be required. The solid findings of this study provide a foundation for further developing drugs targeting BRCA1/2 in lung cancer therapy.

We would like to express our sincere gratitude for your thoughtful and constructive comments on our manuscript. We carefully considered each comment from these two reviewers and revised the manuscript accordingly. Below, we provided a point-by-point response to each comment.

**Reviewer #1 (Public review):**
Summary:Liao et al. performed a large-scale integrative analysis to explore the function of two cancer genes (BRCA1 and BRCA2) in lung cancer, which is one of the cancers with an extremely high mortality rate. The detailed genetic analysis demonstrated new roles of BRCA1/2 in causing the tumor microenvironment in lung cancer. In particular, the discovery of different mechanisms of BRCA1 and BRCA2 provides an essential foundation for developing drugs that target BRCA1 or BRCA2 in lung cancer therapy.Strengths:(1) This study leveraged large-scale genomic and transcriptomic datasets to investigate the prognostic implications of BRCA1/2 mutations in LUAD patients (~2,000 samples). The datasets range from genomics to single-cell RNA-seq to scTCR-seq.(2) In particular, the scTCR-seq offers a powerful approach for understanding T cell diversity, clonal expansion, and antigen-specific immune responses. Leveraging these data, this study found that BRCA1 mutations were associated with CD8+ Trm expansion, whereas BRCA2 mutations were linked to tumor CD4+ Trm expansion and peripheral T/NK cell cytotoxicity.(3) This study also performed a comprehensive analysis of genomic variation, gene expression, and clinical data from the TCGA program, which provides an independent validation of the findings from LUAD patients newly collected in this study.(4) This study provides an exemplary integration analysis using both computational biology and wet bench experiments. The experimental testing in the A549 cell line further supports the robustness of the computational analysis.(5) The findings of this study offer a comprehensive view of the molecular mechanisms underlying BRCA1 and BRCA2 mutations in LUAD. BRCA1 and BRCA2 are two well-known cancer-related genes in multiple cancers. However, their role in shaping the tumor microenvironment, particularly in lung cancer, is largely unknown.(6) By focusing on PD-L1-negative LUAD patients, this study demonstrated the molecular mechanisms underlying resistance to immune therapy. These new insights highlight new opportunities for personalized therapeutic strategies to BRCA-driven tumors. For example, they found histone deacetylase (HDAC) inhibitors consistently downregulated 4-R genes in A549 cells.(7) The deposition of raw single-cell sequencing (including scRNA-seq and scTCR-seq) data will provide an essential data resource for further discovery in this field.Weaknesses:(1) The finding of histone deacetylase (HDAC) inhibitors suggests the potential roles of epigenetic regulation in lung cancer. It would be interesting to explore epigenetic changes in LUAD patients in the future.

Thank you for your insightful comment. We fully agree that the specific situation of epigenetic dysregulation in LUAD needs to be explored. We believe that future investigations utilizing clinical specimens and animal models to map histone acetylation patterns and DNA methylation profiles were crucial for identifying novel biomarkers and therapeutic targets unique to LUAD.

(2) For some methods, more detailed information is needed.

This is a valid point. We agree that additional details regarding are necessary for clarity and reproducibility. We have expanded these method details in the revised manuscript.

(3) There are grammar issues in the text that need to be fixed.

We apologize for our irregular use of grammar. In the revised manuscript, we carefully checked the grammar and make corrections.

(4) Some text in the figures is not labeled well.

We appreciate the reviewers' comments. We have added labels to the revised version of the figures.

**Reviewer #2 (Public review):**
Summary:This study investigates the impact of BRCA1/2 mutations on immunotherapy in lung adenocarcinoma using multi-omics approaches. The work highlights distinct roles of BRCA1 and BRCA2 mutations in shaping immune-related processes, and is logically structured with clearly presented analyses. However, the conclusions rely primarily on descriptive computational analyses and would benefit from additional immunological validation.Strengths:By integrating public datasets with in-house data, this study examines the impact of BRCA1/2 mutations on immunotherapy in lung adenocarcinoma from multiple perspectives using multi-omics approaches. The analyses are diverse in scope, with a clear overall logic and a well-organized structure.Weaknesses:The study is largely descriptive and would benefit from additional immunological experiments or validation using in vivo models. The fact that the BRCA1 and BRCA2 samples were each derived from a single patient also limits the robustness of the conclusions.

Thank you for this excellent suggestion. In the revised manuscript, we supplemented the additional immunological experiments and validation based on pathological tissue sections of lung adenocarcinoma patients. In addition, we elaborated on the limitations of our study in the Discussion section and provided reasonable explanations.

**Recommendations for the authors:**

**Reviewer #1 (Recommendations for the authors):**
(1) The abstract includes a lot of abbreviations, which makes it difficult to follow. For example, "IFN" is not defined. And "HRR" is defined but used only once in the abstract. This issue also appears in other parts, such as "OAK" on page 5, line 114; "DFS" on page 15, line 398; and "DSBs" on page 20, line 558. Please try to avoid unnecessary abbreviations.

Thank you for highlighting this. We have revised the manuscript to minimize the use of abbreviations. Specifically, we have now defined all necessary abbreviations upon first mention (including 'IFN') and have removed or spelled out those used infrequently to ensure the text flows more smoothly for the reader.

(2) Page 5, line 129, what data type is used in this part analysis?

We apologize for our negligence. The whole exome sequencing data used here has been added in the revised manuscript.

Materials and methods, page 6, lines 131-132: “The raw reads (fastq) of whole exome sequencing were pre-processed and trimmed with fastp (Version: 0.23.4) based on default parameters.”

(3) Page 6, line 138, Add citation for ANNOVAR.

Thank you for your suggestion. We have added a citation for ANNOVAR in the revised manuscript.

(4) Page 8, line 211, what cutoff is used to define the significant makers?

Thank you for your insightful comment. We provided the cutoff used to define significant markers.

Materials and methods, page 8, lines 213-215: “Differential expression genes for specific clusters were identified using the “FindMarkers” function, with a threshold of |avg_log2FC| ≥ 0.5 and adjusted P-value ≤ 0.01.”

(5) Page 11, line 276, HEK293T is not a lung cancer cell line. It would be better to label the details of this cell line.

Thank you for your correction. We have now clarified HEK293T in the text by stating: 'human embryonic kidney cell line HEK293T'.

Materials and methods, page 11, lines 277-278: “The human lung cancer cell line A549 (#SCSP-503) and the human embryonic kidney cell line HEK293T (#SCSP-502) were purchased from the Type Culture Collection of the Chinese Academy of Sciences, China.”

(6) Page 16, line 415, what samples and how many individuals were used for the exome sequencing?

We agree that specifying the sample set is crucial. The exome sequencing was conducted on 2 individuals (four samples). The samples used were tumor tissues (2 samples) and matched blood (2 samples). This information has been clarified in the revised manuscript.

Results section, page 16, lines 415-416: “Exome sequencing was performed on four samples from two individuals: two tumor tissues and two matched blood samples.”

(7) Page 17, line 468, Replace "Differently" with "In contrast" (more appropriate for scientific writing).

Thank you for pointing this out. We agree that "In contrast" is more appropriate for scientific writing. Accordingly, we have replaced "Differently" with "In contrast" in this sentence (Results section, page 18, line 483).

(8) Page 18, line 489, what is HMG?

Thank you for pointing this out. HMG stands for High Mobility Group. We have clarified this by writing out the full term upon first mention in the manuscript (Results section, page 19, line 503).

(9) Page 19, line 527, check the grammar for this sentence.

We appreciate your careful reading. We have carefully rephrased this sentence to ensure clarity and grammatical accuracy.

Results section, page 20, line 540: “Based on pseudotime order, we divided trajectories into 10 bins and analyze the activity changes of related features.”

(10) Page 20, line 541-546. It would be better to split this long sentence into smaller ones.

Thank you for your insightful comment. We have revised the text, splitting the long sentence into smaller ones for better clarity.

Results section, page 20, lines 554-559: “MHC class I and II molecules showed increased activity in late pseudotime in *BRCA1*- and *BRCA2*-mutant cells, respectively (Fig. 4G-I). This pattern was also reflected in the cell density analysis (Fig. 4J). Furthermore, CD8^+^ Tcm and Th1 signatures exhibited higher activity in late pseudotime in *BRCA1*- and *BRCA2*-mutant cells, respectively (Fig. S5F-G). These findings suggest a differential association with CD8^+^ versus CD4^+^ T cell engagement.”

(11) Page 20, line 550, remove "." after "of".

Thank you for catching this. We have removed it (Results section, Page 21, line 563).

(12) Page 22, line 592, what is "LME"?

Thank you for pointing this out. "LME" was indeed redundant in the original manuscript, so we have removed it in the revised version (Results section, Page 22, lines 607-609).

(13) Page 24, line 674, Replace "suggest" with "suggested"?

We apologize for our negligence. In the revised manuscript, we have replaced "suggest" with "suggested" (Results section, Page 25, lines 691-693).

(14) Page 35, Figure 1I, Use "B cells" instead of "B".

Thank you for your detailed review. We have changed to the appropriate label in Figure 1I.

(15) Page 36, Figure 2H, the statistics and p-value are needed to show.

Thank you for your suggestion. We have added the statistical analysis for Figure 2H, and the p-values were indicated in the revised Figure.

Special thanks to you for your kind comments.

**Reviewer #2 (Recommendations for the authors):**
Major:(1) Line 44. In the Introduction section, a brief description of the prevalence of HRD or BRCA1/2 mutations in lung cancer patients should be included to highlight the significance of the study.

This is an excellent suggestion. We revised the Introduction section (page 3, lines 61-64) to include a brief overview of the prevalence of *BRCA1/2* mutations specifically in lung cancer patients. We believe this addition will strengthen the background for readers.

Introduction section, page 3, lines 61-64: “Among the key genetic mutations that drive LUAD, *BRCA1* and *BRCA2* mutations (with prevalence rates of approximately 4% and 5%, respectively) have been increasingly implicated in the pathogenesis and progression of lung cancer [9, 13].”

(2) Line 302-355. There are relatively serious grammatical issues, and substantial revision of the text is recommended.

We acknowledge the grammatical issues in the original text. We have now carefully revised the Materials and methods section of the manuscript (pages 11-14, lines 277-358) to correct these issues and improve the overall readability. We believe the revised version is significantly improved.

(3) Line 375. The Results section lacks detailed information on the specific BRCA1/BRCA2 mutations and data explaining how these mutations lead to functional alterations of BRCA1/2.

Thank you for your insightful comment. In the revised manuscript, we added the amino acid changes caused by the specific *BRCA1/BRCA2* mutation sites and expand the text to discuss the predicted and known pathogenic mechanisms of these variants (Results section, page 16, lines 420-433).

Results section, page 16, lines 420-433: “Exome sequencing data show that these two types of tumor tissues harbor somatic nonsynonymous single nucleotide variants (SNV) in *BRCA2* (p.N372H) and *BRCA1* (p.E991G, p.S1566G, p.K1136R, p.P824L, and p.Y809H), respectively (Table S1). The *BRCA2* p.N372H variant lies within the BRC3 or BRC4 motifs critical for RAD51 binding. It may alter binding affinity, impair high-fidelity homologous recombination repair, and promote genomic instability [39-41]. In *BRCA1*, mutations are distributed across two key functional domains: the Coiled-Coil domain (e.g., p.E991G, p.Y809H, p.P824L) and the BRCT domain (e.g., p.K1136R, p.S1566G). Coiled-Coil mutations disrupt BRCA1-PALB2-BRCA2 complex assembly, impairing localization to DNA damage sites and subsequent *RAD51* recruitment; BRCT domain mutations compromise phospho-protein recognition and G2/M checkpoint control, leading to defective DNA damage response and unchecked proliferation of damaged cells [42-44]. Together, these defects promote the accumulation of genomic scars and chromosomal instability.”

(4) Line 492-498. Changes in genes associated with BRCA1 and BRCA2 mutations should be validated by immunofluorescence.

Thank you for your insightful comment. Immunofluorescence would provide valuable orthogonal validation of the protein-level consequences of these mutations. To address this, we obtained pathological tissue sections from patients carrying *BRCA1/2* mutations and performed immunofluorescence staining for *S100A10*, a risk gene associated with *BRCA1* mutations. We found that *S100A10* was upregulated in *BRCA1*-mutated tumor tissue compared to adjacent non-cancerous tissue.

Results section, page 24, lines 673-675: “Immunofluorescence experiments on patient tissue sections revealed that *S100A10* was upregulated in *BRCA1*-mutated tumor tissue relative to adjacent non-cancerous tissue (Fig. S11D-E).”

(5) Line 538. Although both BRCA1 and BRCA2 deficiencies impair DNA damage repair, BRCA1, but not BRCA2, activates the cGAS-STING pathway. This is a particularly interesting observation and should be validated by immunofluorescence experiments.

Thank you for highlighting this observation. To address this, we conducted immunofluorescence experiments to quantify STING, the key protein of cGAS-STING pathway, in *BRCA1*- and *BRCA2*-deficient tissues to confirm this phenotype. We have included these results in the revised manuscript.

Results section, page 21, lines 578-584: “Furthermore, our results revealed that *BRCA1*-mutant tumors showed higher activity of cGAS-STING signaling and STING mediated induction of host immune responses compared to *BRCA2*-mutant tumors (Fig. 5G and Fig. S6F). Also, cGAS-STING signaling gens, including *cGAS*, *STING1*, and downstream factors *STAT1* and *CCL5*, were upregulated in *BRCA1*-mutant tumor cells (Fig. 5H). This observation was validated through immunofluorescence staining experiments on patient tumor tissue sections (Fig. 5I-J).”

(6) Line 599. "CD8+ Trm cells were more abundant in BRCA1-mutant sample, whereas CD4+ Trm cells were higher in BRCA2-mutant sample". This part is also recommended to be validated using immunofluorescence or more rigorous flow cytometry analyses.

We sincerely appreciate this insightful suggestion. To address this, we performed immunofluorescence staining to quantify the abundance of CD8^+^ and CD4^+^ Trm cells in *BRCA1*- and *BRCA2*-mutant tissues. We have included these results in the revised manuscript.

Results section, page 22, lines 614-617: We identified two tissue-resident memory T cell (Trm) subsets, CD8^+^ Trm and CD4^+^ Trm, both predominantly derived from tumor tissues (Fig. 6B). “Interestingly, our analysis revealed that CD8^+^ Trm cells were more abundant in *BRCA1*-mutant tumor, whereas CD4^+^ Trm cells were more abundant in *BRCA2*-mutant tumor (Fig. 6B-D, Fig. S7D, and Fig. S8A-B).”

(7) Line 643-676. The authors identified four risk genes associated with BRCA1 mutations-S100A10, LDHA, MYL12A, and GAPDH; however, MYL12A was not validated in the subsequent in vitro experiments. The authors state that "S100A10 can promote cancer metastasis by recruiting MDSC cells, and increased LDHA activity contributes to tumor immune escape." However, because immune cells were not included in the in vitro assays, these results instead suggest that these genes may directly suppress tumor cell proliferation.

We thank the reviewer for this insightful observation. Our intention was not to suggest that the reduction in proliferation observed in our in vitro assays was caused by the disruption of immune cell recruitment or immune escape pathways. As the reviewer correctly points out, those mechanisms are irrelevant in a system lacking immune cells. Our results showing that "Knockdown of *S100A10*, *LDHA*, and *GAPDH* reduced LUAD cell proliferation in vitro (Fig. 7D-E)" strongly suggest a direct, cell-autonomous role for these genes in regulating LUAD cell growth. For the *MYL12A* gene, the existing study have shown that *BRCA1* transcriptionally regulates this gene involved in breast tumorigenesis (PMID: 12032322). In view of the characteristics of *MYL12A* in lung cancer, we will conduct in-depth in vitro and in vivo validation experiments in future studies.

(8) Line 677. The authors should emphasize the limitations arising from the small sample size and the lack of in vivo validation models in the Discussion section.

Thank you for highlighting these important limitations. We agree that the small sample size and the lack of in vivo validation are significant limitations of the current study. We have explicitly addressed these points in the Discussion section (page 27, lines 740-750) to ensure the interpretation of our data is appropriately qualified and to provide transparency regarding the scope of our conclusions.

Discussion section, page 27, lines 740-750: “Although we included both tumor tissues and matched paracancerous and blood samples, the sample size remains modest, which may limit the statistical power and generalizability of our findings. Therefore, our results should be interpreted as preliminary, and further studies with larger, independent cohorts are required to validate these observations. Single-cell RNA-seq and TCR-seq analyses in this study provide high-resolution insights into the cellular and clonal dynamics of the TME, the functional validation of key mechanisms remains largely correlative. While our in vitro experiments provide valuable mechanistic insight, the lack of in vivo validation, which cannot fully recapitulate the complex TME. Future studies utilizing murine models or patient-derived organoids are essential to establish causal relationships and elucidate the underlying molecular pathways.”

Minor:(1) Line 163: cell/μl should be corrected to cells/μL.

Thank you for catching this. We have corrected it in the revised manuscript (Methods section, page 7, line 165).

(2) Line 388: Please clarify how the HRD score, tumor mutation burden, and neoantigen load were calculated.

We thank the reviewer for this request for clarification. In the revised manuscript, we have expanded the Methods section (page 5, lines 117-121) to provide a detailed description of how these metrics were calculated. HRD score was calculated as the unweighted sum of loss of heterozygosity (LOH), telomeric allelic imbalance (TAI), and large-scale state transitions (LST). Tumor mutation burden (TMB) was defined as the total number of somatic nonsynonymous mutations per megabase of the exome captured by the sequencing panel. Neoantigen load was predicted by NetMHCpan using the patient's HLA typing and the identified somatic mutations. The data for these three indicators all obtained from a previous study (PMID: 29628290). We believe these additions provide the necessary transparency and reproducibility for our study.

Methods section, page 5, lines 117-121: The HRD score was determined by summing specific genomic alterations, including loss of heterozygosity (LOH), large-scale state transitions (LST), and telomeric allelic imbalances (TAI). “Tumor mutation burden (TMB) was defined as the total number of somatic nonsynonymous mutations per megabase of the exome captured by the sequencing panel. Neoantigen load was predicted by NetMHCpan using the patient's HLA typing and the identified somatic mutations.”

(3) Line 421: BRCA12 should be corrected to BRCA2.

Thank you for your detailed review. We have revised it.

(4) The order of Figures 7D and 7E should be reversed.

Thank you for your insightful comment. According to your suggestion, we reversed the order of Figures 7D and 7E in the revised manuscript.

Special thanks to you for your kind comments.